# Ski Rental with Distributional Predictions of Unknown Quality

**Qiming Cui** [1]  **Michael Dinitz** [2]

## Abstract

We revisit the central online problem of *ski rental* in the "algorithms with predictions" framework from the point of view of *distributional* predictions. If we are given as a prediction a *distribution* $\hat{p}$ over the ski days, and the true number of ski days comes from some (unknown) distribution $p$, then we show as our main result that there is an algorithm with expected cost at most $OPT + O\left(\min\left(\max(\eta, 1) \cdot \sqrt{b},\ b \log b\right)\right)$, where $OPT$ is the expected cost of the optimal policy for the true distribution $p$, $b$ is the cost of buying, and $\eta$ is the Earth Mover's (Wasserstein-1) distance between $p$ and $\hat{p}$. An implication of this bound is that our algorithm has *consistency* $O(\sqrt{b})$ (additive loss when the prediction error is 0) and *robustness* $O(b \log b)$ (additive loss when the prediction error is arbitrarily large). Moreover, we do not need to assume that we know (or have any bound on) the prediction error $\eta$, in contrast with previous work in robust optimization which assumes that we know this error. We also complement this upper bound with a variety of lower bounds showing that it is essentially tight: not only can the consistency/robustness tradeoff not be improved, but our particular loss function cannot be meaningfully improved.

## 1. Introduction

The paradigm of algorithms with predictions has emerged as an important framework for enhancing decision-making under uncertainty, making it possible to combine the benefits of machine learning (ML) with traditional worst-case algorithmic guarantees. The goal is to use advice (presumably from an ML model) in a way that allows us to approach optimal decisions when this advice is accurate (known as a *consistency* guarantee), but also allows us to fall back on worst-case guarantees when the advice is inaccurate (known as a *robustness* guarantee). This allows us to simultaneously get the benefits of ML (extremely good algorithms when the learning accurately reflects the input) and the benefits of traditional algorithms (worst-case performance) without suffering from the downsides of ML (poor performance when the input is dramatically different from the past) or the downsides of traditional algorithms (weak average-case performance due to emphasis on the worst-case).

The ski rental problem stands as a cornerstone of online algorithms, epitomizing the fundamental rent-or-buy dilemma where an agent must balance per-use costs against irreversible investments without knowledge of the problem horizon. The total number of ski days $T$ is unknown to the skier, and every day the skier must either rent skis (for a cost of 1) or buy skis (for a cost of $b > 1$). If they buy skis, then they no longer have to rent, since they now own. Clearly if the number of ski days $T$ is at least $b$ then buying at the beginning of the season is optimal, while if the $T < b$ then renting every day is optimal. But without knowing $T$, what should we do? A classical result of Karlin et al. (1986) is that the optimal deterministic strategy achieves a competitive ratio of 2 by renting until day $b - 1$ and then buying (i.e., this strategy never pays more than twice the optimal strategy knowing $T$).

Due to its fundamental nature and importance, ski rental was one of the first settings to be looked at in the context of predictions: Purohit et al. (2018) showed how to use a prediction $\hat{T}$ of the skiing duration $T$ in a way that performs well when the prediction is accurate, but still has reasonable worst-case bounds if the prediction is inaccurate. An important feature of Purohit et al. (2018), as well as most (but not all) other work on algorithms with predictions, is that the prediction is fundamentally a *point*: we are given a prediction of the number of days that we will be skiing. But predictions from ML models are inherently stochastic, and moreover since we want to do well in settings where we expect ML to do well (i.e., when the future looks like the past), we might expect $T$ to also be drawn from a distribution (hopefully one that is related to the past empirical distribution). So we might expect both our prediction and

[1]Department of Applied Mathematics and Statistics, Johns Hopkins University, Baltimore, Maryland, USA [2]Department of Computer Science, Johns Hopkins University, Baltimore, Maryland, USA. Correspondence to: Qiming Cui <qcui6@jhu.edu>, Michael Dinitz <mdinitz@cs.jhu.edu>.

*Proceedings of the 43$^{rd}$ International Conference on Machine Learning*, Seoul, South Korea. PMLR 306, 2026. Copyright 2026 by the author(s).

the truth to really be *distributional*.[1] While one could of course sample or take the MLE of the distribution to turn it into a point, it is not hard to see that this can be extremely lossy. For example, suppose the true and predicted distributions are both bimodal, with one mode on short seasons and one mode on long seasons. Sampling a single point prediction collapses the distribution to one of the two modes and discards the other. If the sampled point comes from one mode while the realized season comes from the other, the point-prediction algorithm can make a decision tailored to the wrong regime, leading it to buy far too early or rent for far too long. Later in the paper, we illustrate this behavior in a small synthetic experiment.

Motivated by this thought process, we study ski rental with *distributional predictions*: the true value $T$ is drawn from some distribution $p$, and we are given a prediction $\hat{p}$. Our goal is, as always, to achieve good performance when $\hat{p}$ and $p$ are close (consistency) while still providing worst-case guarantees when $p$ and $\hat{p}$ are arbitrarily far apart. We discuss our results in Section 2, but at a high level, we are able to do exactly that: we design an algorithm which uses a distributional prediction in an essentially optimal way to achieve small additive loss when $p$ and $\hat{p}$ are close, while also being robust to incorrect predictions.

We note that we are not the first to consider such a setting. Most notably, ski rental with some type of distributional prediction was also studied by Diakonikolas et al. (2021) and by Besbes et al. (2025). We discuss these and other related work in Section 2.3, but at a high level, Diakonikolas et al. (2021) focuses on the sample-complexity question of actually learning the true distribution, while Besbes et al. (2025) looks at almost the same setting as us except from a robust optimization point of view: they assume that they know the distance between $p$ and $\hat{p}$ (or at least a good bound on it). This enables them to get significantly stronger upper bounds, but is in contrast to essentially all other work in the algorithms with predictions framework, which does not typically assume such knowledge. Moreover, they explicitly raise our setting as an open question: "A natural question would be to understand the best achievable performance without knowledge of $\epsilon$ [distance between $p$ and $\hat{p}$]" (Besbes et al., 2025).

## 2. Our Results and Contributions

### 2.1. Problem Setup and Definitions

Before we can state our results, we need to introduce the problem setup and some basic definitions more formally. We normalize so the rental cost is one unit cost per day and

---

[1]Clearly a point distribution is a distribution, and so distributional predictions and truths strictly generalize the traditional point case.

the buying cost is $b$. An adversary chooses a distribution $p$ over $[N]$ for some large finite value $N$, and the total number of ski days $T$ is drawn from $p$. We are given as a prediction a distribution $\hat{p}$ over $[N]$, and our goal is to design an algorithm (or policy) which outputs a decision for every day $i$ whether to rent or buy (assuming that we are still skiing on day $i$). Note that once the policy decides to buy, there are no longer any decisions for it to make. A policy can be deterministic or randomized; our algorithms will be deterministic, but our lower bounds will apply to both randomized and deterministic policies. Clearly any deterministic policy is of the form "rent for $K$ days, then buy on day $K + 1$", and we denote such a policy as $A_K$. To simplify notation, we also let $A_0$ denote the policy of "buy at the beginning" (i.e., after 0 days) and we let $A_\infty$ denote the policy of renting forever (until the process ends).

Given a prediction $\hat{p}$ with some unknown truth $p$, we design an algorithm $ALG(\hat{p})$. We use the notation $ALG_p(\hat{p})$ to refer to running algorithm $ALG$ obtained from $\hat{p}$ when the true distribution is $p$. If the true number of ski days is $T$ (possibly drawn from a distribution $p$), we abuse notation and let $ALG_T(\hat{p})$ denote running the algorithm with true value $T$. Since our algorithm naturally depends on $\hat{p}$, we use $ALG$ or $ALG(\hat{p})$ interchangeably. Let $c(\cdot)$ be the (expected) cost function, so $c(ALG_T)$ is the expected cost for $ALG$ with a point truth $T$ and $c(ALG_p) = \mathbb{E}_{T \sim p}[c(ALG_T)]$. Moreover, each policy has a corresponding expected cost:

$$c((A_K)_p) = \sum_{t \le K} t\, p_t \;+\; \sum_{t > K} (K + b)\, p_t, \qquad \text{for } K \ge 0,$$

$$c((A_\infty)_p) = \sum_t t\, p_t.$$

The optimal policy for $p$ (which we denote by $OPT_p$) is the policy that minimizes this expected cost, i.e., $c(OPT_p) = \arg\min_{K \in \{0,\dots,N\} \cup \{\infty\}} c((A_K)_p)$.

In order to claim that our algorithm does well when $\hat{p}$ is "close" to $p$, we need some definition of distance between distributions. We will use Earth Mover's distance ($EMD$) (also known as the *Wasserstein-1 metric*) to measure the error between the predicted distribution and the truth. This distance is formally defined as $EMD(P, Q) = \inf_{\gamma \in \Pi(P,Q)} \mathbb{E}_{(x,y) \sim \gamma}[\|x - y\|]$ where $\Pi(P, Q)$ denotes the set of joint distributions (or couplings) with marginals $P$ and $Q$. Informally, this is equivalent to the minimum total mass movement necessary to turn distribution $P$ into $Q$. Slightly more carefully, a *transport plan* $\pi(x, y)$ specifies the amount of mass transported from point $x$ to point $y$ (with $\pi(x, y) \ge 0$ for all $x, y$), and satisfies the marginal constraints $\sum_y \pi(x, y) = P(x)$ and $\sum_x \pi(x, y) = Q(y)$. The *optimal transport plan* is the plan that minimizes $\sum_{x,y} \pi(x, y)|x - y|$, and this minimum cost is $EMD(P, Q)$. For brevity, we also use $EMD$ when the chosen distributions are clear from context.

**Benchmark: Expected Optimal Policy Cost.** Our aim is to design an algorithm $ALG$ with small additive loss compared to the optimal policy for $p$. More carefully, the *additive loss* of an algorithm $ALG$ on distribution $p$ is $\text{diff}_p := c(ALG_p) - c(OPT_p)$, and we want to minimize this quantity.

Note that we choose our benchmark to be the expected cost of the optimal policy, rather than the optimum in hindsight. If $p$ is a point distribution (i.e., the adversary just selects a fixed number of days $T$) then these are the same thing, but in general they can be quite different. While it may not be clear a priori which is the better benchmark, since any algorithm is actually a policy, a "fair" comparison is to the best policy; this allows us to quantify the loss from an inaccurate prediction, rather than combining the loss from an inaccurate prediction with inherent loss due to stochasticity.

Moreover, it is not hard to show that it is simply not possible to achieve good performance compared to the optimum in hindsight: our additive error must be $\Omega(b)$. Consider the following example: $p_{\frac{b}{2}} = \frac{1}{2}$ and $p_{2b} = \frac{1}{2}$. Then it is not hard to see that both $A_0$ and $A_{b/2}$ are the optimal policies for $p$, each of which has expected cost $b$. On the other hand, the expected optimum in hindsight has cost $\frac{1}{2} \cdot \frac{b}{2} + \frac{1}{2} \cdot b = \frac{3}{4}b$. So even if we assume that we know $p$ precisely, our additive loss compared to the optimum in hindsight is $\Omega(b)$. By using the optimal policy as our baseline (a more fair comparison) we will be able to get additive loss that is sublinear in $b$.

### 2.2. Our Results

**Main Upper Bound.** We begin with our main result.

**Theorem 1.** *There is an algorithm $ALG$ which takes as input a distributional prediction $\hat{p}$ and has*

$$c(ALG_p) - c(OPT_p) \leq$$
$$O\left(\min\left(\sqrt{b} \cdot \max\left(EMD(\hat{p}, p), 1\right), \, b\log b\right)\right).$$

To interpret this bound, first consider the case where $p = \hat{p}$. In this case the additive loss $c(ALG_p) - c(OPT_p)$ will be $O(\sqrt{b})$. So our algorithm has *consistency* of $O(\sqrt{b})$. As the prediction gets increasingly inaccurate (i.e., $EMD(\hat{p}, p)$ grows), our algorithm has additive loss that stays at $O(\sqrt{b})$ until $EMD(\hat{p}, p) > 1$, and then grows linearly as $O(\sqrt{b} \cdot EMD(\hat{p}, p))$ until $EMD(\hat{p}, p)$ reaches $\Theta(\sqrt{b}\log b)$, after which the loss stays at $O(b\log b)$.

The most obviously comparable previous work is Besbes et al. (2025). In their setting they assume that $EMD(\hat{p}, p)$ is known to the algorithm, although of course $p$ itself is not known. In this setting they achieve an improved upper bound of $O\left(\sqrt{b \cdot EMD(\hat{p}, p)} + EMD(\hat{p}, p)\right)$. Moreover, in their setting robustness becomes essentially trivial:

we can always get worst-case additive loss of $b$ by just checking whether the guarantee of the algorithm is larger than $b$, and if it is, switching the algorithm to just buy on the first day. In our setting, though, robustness becomes a key challenge. It is obviously important, since if we just run a non-robust algorithm blindly without knowing $EMD(\hat{p}, p)$ we might incur arbitrarily large additive loss. And yet it is not obvious how to achieve robustness without harming performance in the low $EMD(\hat{p}, p)$ regime (consistency).

To prove Theorem 1, we first design a non-robust base algorithm which has our desired loss except without robustness, i.e., it has loss $O\left(\min\left(\sqrt{b} \cdot \max\left(EMD(\hat{p}, p), 1\right)\right)\right)$. Our algorithm is quite simple: we compute the optimal policy for the prediction $\hat{p}$, and then delay buying by an additional $\sqrt{b}$ days. It turns out that, even though Besbes et al. (2025) focused on the fundamentally different setting of known $EMD(\hat{p}, p)$ and do not provide any lemmas or theorems which directly apply to this algorithm, our bound can be derived from equation (E-28) in the online appendix of their paper. Nevertheless, we believe that our analysis is somewhat more direct and intuitive since it directly uses properties of the optimal transport plan, so we include it for completeness (and since it was developed independently). We give our full proof of this property in Appendix C.

The particular choice of a $\sqrt{b}$ delay comes from balancing two sources of error in the transport-based analysis. To see the intuition, suppose more generally that we delayed the optimal policy for $\hat{p}$ by $D$ days. The only probability mass that can cause a large change in cost is mass that moves far enough under the optimal transport plan to cross this additional $D$-day delay. Since the total transportation cost is $EMD(\hat{p}, p)$, the amount of such mass is at most roughly $EMD(\hat{p}, p)/D$. Each unit of such mass can change the rental cost by at most $b$, in our required upper bound the coefficient becomes $b/D$. On the other hand, in the transport plan there is also the part where the moving distance contributes directly; this is bounded by $D$ times the transported mass, and hence by $D$. The choice $D = \sqrt{b}$ then gives the best tradeoff, which explains the shift used by the algorithm.

With the base algorithm in hand, the question is now how to achieve robustness. We do this by adding a truncation condition based on the tail probability mass in $\hat{p}$. Slightly more carefully, we compute the first day $U$ where the total probability mass in $\hat{p}$ on days larger than $U$ is at most $1/\sqrt{b}$. If $U$ is smaller than the optimal policy threshold $\hat{K}$ for $\hat{p}$, then we switch to instead use $A_{U+\sqrt{b}}$ (buy on day $U + \sqrt{b} + 1$) rather than use $A_{\hat{K}+\sqrt{b}}$ (buy on day $\hat{K} + \sqrt{b} + 1$) as our base algorithm would.

We need to show that this modification does not increase the loss when $EMD(\hat{p}, p)$ is small, and also need to show

that it actually implies robustness of $O(b \log b)$. For the former, this boils down to showing that if the truncation actually modifies the algorithm, it must be because the optimal buying time for $\hat{p}$ occurs when there is very little probability mass left, and thus this is a low-probability event that only affect the loss by a constant factor. While this is only "low probability" with respect to the prediction $\hat{p}$ and not the truth $p$, we only need to apply this to the setting when $EMD(\hat{p}, p) \leq \sqrt{b} \log b$, and so we can translate between $p$ and $\hat{p}$ without too much extra loss.

For the latter, if $\hat{K} < U$, by the fact that the predicted tail mass at time $\hat{K}$ is still non-negligible (because $U$ has not been reached yet), and arguing that the optimality of $A_{\hat{K}}$ forces the predicted tail mass to decrease quickly enough, we are able to show that $\hat{K}$ cannot be too large, which is itself enough to imply the desired robustness. If $U \leq \hat{K}$, the optimality of buying at $\hat{K}$ imposes a strong structural constraint on how the predicted tail can behave: intuitively, to keep $\hat{K}$ optimal, the predicted tail must lose some fraction of its remaining mass over a bounded time window. Thus, we are able to show that the time required for the predicted tail to drop to the level defining $U$ is at most $O(b \log b)$. The full proof of Theorem 1 can be found in Section 3.

**Further Robustness.** Another interpretation of Theorem 1 is that we can get robustness of $O(b \log b)$ *for free*. We prove later that even if one does not care about robustness, no algorithm can have additive loss better than our $O\left(\sqrt{b} \cdot \max\left(EMD(\hat{p}, p), 1\right)\right)$ (with some caveats; see Theorem 5). We manage to achieve this bound while also achieving robustness $O(b \log b)$.

But what if we want a worst-case guarantee below $O(b \log b)$? To obtain a stronger robustness guarantee, we introduce an additional robustification step that interpolates between Algorithm 1 and the classical worst-case algorithm; this interpolation improves worst-case performance, but may come at the cost of degraded performance in the low-error regime. To do this, we follow the lead of Purohit et al. (2018) in the point prediction setting: we design an algorithm which takes an additional parameter $\lambda$ corresponding to how much we "trust" our prediction, and allows us to smoothly interpolate between the fully trusted setting ($\lambda = 0$) and the fully untrusted setting ($\lambda = 1$).

**Theorem 2.** *There is an algorithm ALG which takes as input a parameter $\lambda \in [0, 1]$ and a distributional prediction $\hat{p}$, and has expected cost*

$$c(ALG_p) \leq \min\left\{(1 + \lambda)\left(c(OPT_p) + \mathcal{B}_{\hat{p}, p}\right),\right.$$
$$\left.\left(1 + \frac{1}{\lambda}\right)c(OPT_p)\right\},$$

*where $\mathcal{B}_{\hat{p}, p} = O\left(\min\left(\sqrt{b} \max\left(EMD(\hat{p}, p), 1\right), b \log b\right)\right)$.*

Note that when $\lambda = 0$ the minimum is achieved by the first term and exactly matches the bound of Theorem 1, while if $\lambda = 1$ then the minimum is achieved by the second term and the bound can be rewritten as $c(ALG_p)/c(OPT_p) \leq 2$, generalizing and recovering the classical 2-competitive algorithm. Choosing $\lambda \in (0, 1)$ allows us to interpolate smoothly between these extremes.

This algorithm, as well as the proof of Theorem 2 can be found in Appendix A. The main idea behind this further robustification step is to "push" our prediction-based algorithm closer to the traditional worst-case algorithm (renting until day $b - 1$ before buying): while the algorithm from Theorem 1 suggests buying on day $D$, one can potentially delay the purchase to a later time $D' < b$ if $D < b$, or make an early purchase on a preceding day $D'' > b$ if $D > b$.

**Lower Bounds.** Despite the complexity of the upper bound in Theorem 1, we provide a collection of lower bounds showing that it is tight in a variety of ways. To begin, recall that Purohit et al. (2018) studied the ski rental problem in the setting where the true number of ski days is $T$ and we are given a prediction $\hat{T}$. One of their first results is an algorithm which has cost $OPT + |T - \hat{T}|$, or in other words, $OPT$ plus the error of the prediction. With that result in mind, a natural goal would be for us to generalize this to the distributional case and give an algorithm with expected cost at most $OPT_p + EMD(\hat{p}, p)$. Unfortunately, as our first lower bound, we show that this is impossible (the proof of this theorem can be found in Appendix B).

**Theorem 3.** *Let $\hat{p}$ be the distributional prediction and $p$ be the unknown true distribution. For any $0 < \epsilon \leq 1$, there is no algorithm (deterministic or randomized) ALG such that*

$$c(ALG_p) - c(OPT_p) \leq O(EMD(\hat{p}, p)) + O(b^{1 - \epsilon}).$$

There is, however, a limited setting in which we can get around this lower bound: if $p$ is a point distribution and $\hat{p}$ is an arbitrary distribution, then simply by sampling a prediction $\hat{T}$ from $\hat{p}$ and using that in the algorithm of Purohit et al. (2018) achieves additive loss of $O(EMD(\hat{p}, p))$. In Appendix D we give a slight improvement by designing a deterministic version of this algorithm.

Theorem 3 implies that any non-trivial algorithm must have loss in which $EMD(\hat{p}, p)$ and $b$ are combined nonlinearly, as in our non-robust upper bound of $O\left(\sqrt{b} \cdot \max(EMD(\hat{p}, p), 1)\right)$. A natural question is whether it is actually necessary to have something like $\max(EMD(\hat{p}, p), 1)$ in this bound. We can show that it is, at least as long as we want the multiplier of $EMD(\hat{p}, p)$ to be $o(b)$ (for the full proof see Appendix B).

**Theorem 4.** *Let $\hat{p}$ be a distributional prediction and $p$ be the unknown true distribution. There is no algorithm ALG*

*(deterministic or randomized) with expected additive loss* $c(ALG_p) - c(OPT_p) \leq o(b) \cdot EMD(\hat{p}, p)$.

So when $\hat{p}$ is extremely close to $p$ (subconstant $EMD(\hat{p}, p)$) we need to allow some extra error, which is precisely what our bound of $\sqrt{b} \cdot \max(EMD(\hat{p}, p), 1)$ does. But now that we have justified the use of $\max(EMD(\hat{p}, p), 1)$, a natural next question is whether the $\sqrt{b}$ term is tight. Our next lower bound implies that it is, by proving a lower bound that precisely matches our upper bound under the restriction that the bound be of the form $f(b) \cdot \max(EMD(\hat{p}, p), 1)$.

**Theorem 5.** *Let $\hat{p}$ be a distributional prediction and $p$ be the unknown true distribution. There is no algorithm $ALG$ (deterministic or randomized) with $c(ALG_p) - c(OPT_p) \leq o\left(\sqrt{b}\right) \cdot \max(EMD(\hat{p}, p), 1)$.*

The previous lower bounds are about the non-robust version of the algorithm. For our final lower bound, we show that our main algorithm (with the robustness guarantee) is optimal from the perspective of robustness vs. consistency. As mentioned, Theorem 1 implies that our algorithm has additive loss $O(\sqrt{b})$ when $EMD(\hat{p}, p) = 0$ (consistency) and has additive loss $O(b \log b)$ when $EMD(\hat{p}, p)$ is arbitrarily large (robustness). We show that this is pareto-optimal: any algorithm with this consistency (or better) must have this robustness (or worse). The proof of this theorem can be found in Section 4.

**Theorem 6.** *For any deterministic or randomized algorithm $ALG$, if $c(ALG_p) - c(OPT_p) \leq O\left(\sqrt{b}\right)$ when $EMD(\hat{p}, p) = 0$, then $c(ALG_p) - c(OPT_p) \geq \Omega(b \log b)$ for large enough prediction error $EMD(\hat{p}, p)$.*

**Experiments.** While our main contribution is theoretical, we provide some small experiments as a sanity check. We will discuss one experiment here, and another in Appendix E. For comparison, a natural baseline is to sample a point from the predicted distribution and then run an existing point-prediction algorithm. This approach can perform poorly on genuinely bimodal instances because sampling collapses the prediction to a single mode. We now see what happens in this scenario experimentally. In particular, we compare our main algorithm in Theorem 1 with the best known point-prediction algorithm of Purohit et al. (2018). We set $b = 100$ and let the true distribution $p$ satisfy $p_{30} = 1/2$ and $p_{300} = 1/2$. For $d \in \{0, 5, 10\}$, we use the predicted distribution $\hat{p}^{(d)}$ given by $\hat{p}^{(d)}_{30-d} = 1/2$ and $\hat{p}^{(d)}_{300+d} = 1/2$, so that $EMD(p, \hat{p}^{(d)}) = d$. For the point-prediction baseline, we sample a single point $\hat{T} \sim \hat{p}^{(d)}$ and then run the point-prediction algorithm of Purohit et al. (2018), which buys immediately if $\hat{T} \geq b$ and otherwise keeps renting. For each value of $d$, we ran 200,000 Monte Carlo trials, drawing the true season length $T \sim p$ and, for the point-prediction baseline, independently drawing $\hat{T} \sim \hat{p}^{(d)}$. The

*Table 1.* A small bimodal experiment with $b = 100$, true distribution $p_{30} = p_{300} = 1/2$, and predicted distribution $\hat{p}^{(d)}_{30-d} = \hat{p}^{(d)}_{300+d} = 1/2$. Costs of the algorithms are Monte Carlo averages over 200,000 trials. For reference, the optimal policy under the true distribution $p$ has an expected cost 80.0 (theoretical benchmark).

| $EMD(\hat{p}^{(d)}, p)$ | Our Cost | Point-Prediction Cost | $OPT_p$ |
|---|---|---|---|
| 0 | 85.1067 | 132.9766 | 80.0 |
| 5 | 87.3655 | 131.9728 | 80.0 |
| 10 | 90.1374 | 132.5884 | 80.0 |
| 15 | 92.6225 | 132.5493 | 80.0 |

resulting average costs are shown in Table 1. This illustrates the regime where distributional predictions are perhaps the most useful: when the prediction is genuinely bimodal, our algorithm in Theorem 1 is significantly better than sampling from the prediction and then applying the point prediction method and is much closer to the exact optimal policy cost for the true distribution. Further discussion is deferred to Appendix E.

## 2.3. Related Work

As mentioned earlier, since its introduction by the seminal paper of Lykouris & Vassilvitskii (2021), the algorithms with predictions framework has seen an explosion of work. This literature is too vast to summarize, so we instead just point the interested reader to the early survey of Mitzenmacher & Vassilvitskii (2022) and the excellent online list of papers maintained by (Lindermayr & Megow, 2025). The non-ski rental part of this literature that is most closely connected to this paper is the recent work on binary search with distributional predictions (Dinitz et al., 2024), which studied a very similar problem in the context of binary search (how to utilize a distributional prediction as opposed to the previous literature that focused on point predictions).

The ski rental problem is one of the most fundamental online problems, modeling the fundamental "rent-or-buy" question that is at the heart of many online decision-making tasks. It and its variants have been studied extensively, including in the context of algorithms with predictions (Antoniadis et al., 2021; Besbes et al., 2025; Bhattacharya & Das, 2022; Diakonikolas et al., 2021; Purohit et al., 2018; Shen et al., 2025; Shin et al., 2023a;b; Wang et al., 2020). The papers most related to our work are Purohit et al. (2018), Diakonikolas et al. (2021), and Besbes et al. (2025). The state-of-the-art algorithm for the classical ski rental problem with point predictions is due to Purohit et al. (2018), whose work is also a key reference here. They provide a non-robust algorithm for point predictions, as well as a robust version, and our results are heavily inspired by theirs.

Diakonikolas et al. (2021) and Besbes et al. (2025) also

extend to distributional predictions, as we do, but their papers have different focuses and results. Diakonikolas et al. (2021) focus on the *sample-complexity* problem. Rather than assuming that we are given a distribution, they assume that we have *sample access* to a distribution and study how many samples are needed before we can design algorithms with performance comparable to the optimal policy for that distribution. By assuming the distribution belongs to a structured family (e.g., log-concave), they can get strong sample-complexity bounds. But they are not concerned with what happens when we sample from a *different* distribution, and do not provide any bounds that are a function of the distance between the prediction (the distribution they sample from) and the true distribution.

Besbes et al. (2025) also considers a distributional prediction and considers both the low-sample case of Diakonikolas et al. (2021) and the full distributional information case that we consider. However, in line with classical work on robust optimization, they assume that they know $EMD(\hat{p}, p)$. Their setting can be phrased as optimization under an uncertainty set: given a prediction $\hat{p}$ and a value $\eta$ with the promise that $EMD(\hat{p}, p) \leq \eta$, they want to design a policy that is competitive with the optimal policy for $p$. This assumption that they know $EMD(\hat{p}, p)$ allows for very strong results. For example, if $EMD(\hat{p}, p) = 0$ then they can simply run the optimal policy for $p$, since they know that their prediction is accurate. And since they know $EMD(\hat{p}, p)$ they are not concerned with designing robust algorithms (as we do in Theorem 1 and Theorem 2): if $EMD(\hat{p}, p)$ is large then they can run the classical worst-case algorithm from the beginning, and so robustness is trivial.

## 3. Main Algorithm

In this section we focus on Theorem 1, which is our main upper bound: we give an algorithm that not only achieves an additive loss of $O(\sqrt{b} \cdot \max(EMD(\hat{p}, p), 1))$, but guarantees an additive loss bounded by $O(b \log b)$. Section 2.2 provides the high-level ideas for the analysis. Here, we present the main algorithm (Algorithm 1), together with the formal theorem statement and proofs. We assume without loss of generality that $\sqrt{b}$ is an integer; otherwise, we use $\lfloor \sqrt{b} \rfloor$ instead.

Recall the statement of Theorem 1:

**Theorem 1.** *There is an algorithm ALG which takes as input a distributional prediction $\hat{p}$ and has*

$$c(ALG_p) - c(OPT_p) \leq$$
$$O\left(\min\left(\sqrt{b} \cdot \max\left(EMD(\hat{p}, p), 1\right), b \log b\right)\right).$$

The algorithm that we will use to prove Theorem 1 is Algorithm 1, which we gave informally in Section 2.2 so now

present formally. Note that $\hat{K}$ could be $\infty$ in this algorithm (if the optimal policy to $\hat{p}$ is to never buy), but $U$ is always finite.

---

**Algorithm 1** Main algorithm
___
1: **Compute** the optimal policy $A_{\hat{K}}$ for the prediction $\hat{p}$.
2: **Compute** the threshold $U$ such that the remaining tail mass of $\hat{p}$ is at most $1/\sqrt{b}$, i.e.,

$$U = \min\left\{ t : \sum_{\tau > t} \hat{p}_\tau \leq \frac{1}{\sqrt{b}} \right\}.$$

3: **Set** $K^* = \min(\hat{K} + \sqrt{b}, U + \sqrt{b})$.
4: **Run** $A_{K^*}$.

---

The rest of this section is devoted to proving that Algorithm 1 has the bound claimed by Theorem 1. This proof proceeds in two parts. We first bound the additive loss of Algorithm 1 by $O\left(\sqrt{b} \cdot \max(EMD(\hat{p}, p), 1)\right)$ (Theorem 9) and then establish a second bound of $O(b \log b)$ (Theorem 12). The theorem follows by taking the minimum of the two bounds.

We first prove a useful lemma relating tails of distributions.

**Lemma 7.** *Let $p$ and $\hat{p}$ be distributions over $\mathbb{Z}_{\geq 0}$. Then for any $a \geq 0$ and any $s > 0$,*

$$\sum_{t > a + s} p_t \leq \sum_{t > a} \hat{p}_t + \frac{EMD(\hat{p}, p)}{s}.$$

*Proof.* Let $\pi(x, y)$ be an optimal transport plan achieving $EMD(\hat{p}, p)$: formally, let $\pi : [N] \times [N] \to \mathbb{R}_{\geq 0}$ such that $\sum_y \pi(x, y) = \hat{p}_x$, $\sum_x \pi(x, y) = p_y$, and $EMD(\hat{p}, p) = \sum_{x, y} \pi(x, y) |x - y|$. Such a $\pi$ must exist by the definition of $EMD(\hat{p}, p)$.

We decompose the tail mass of $p$ beyond $a + s$ according to the origin index $x$ under $\pi$:

$$\sum_{t > a + s} p_t = \sum_{\substack{y > a + s \\ x > a}} \pi(x, y) + \sum_{\substack{y > a + s \\ x \leq a}} \pi(x, y).$$

The first term is at most $\sum_{t > a} \hat{p}_t$, since it only uses mass originating from indices $x > a$.

For the second term, observe that whenever $y > a + s$ and $x \leq a$, we have $|x - y| > s$. Therefore,

$$\sum_{\substack{y > a + s \\ x \leq a}} \pi(x, y) \leq \frac{1}{s} \sum_{\substack{y > a + s \\ x \leq a}} \pi(x, y) |x - y|$$

$$\leq \frac{1}{s} \sum_{x, y} \pi(x, y) |x - y| \leq \frac{EMD(\hat{p}, p)}{s}.$$

Combining the two bounds yields

$$\sum_{t>a+s} p_t \leq \sum_{t>a} \hat{p}_t + \frac{EMD(\hat{p}, p)}{s}. \qquad \square$$

A crucial first step in proving the non-robust bound will be to bound the loss of the "base algorithm", i.e., the variant of Algorithm 1 that skips step 2 and just sets $K^* = \hat{K} + \sqrt{b}$.

**Theorem 8.** *The base algorithm has expected additive loss*

$$c(ALG_p) - c(OPT_p) \leq O\left(\sqrt{b} \cdot \max\left(EMD(\hat{p}, p), 1\right)\right)$$

As mentioned, this follows directly from equation (E-28) in the online appendix of Besbes et al. (2025) by setting $\delta = -\sqrt{b}$ in that equation. However, we include an independently discovered proof of this theorem in Appendix C, as we believe that it is more intuitive and direct.

With this fact in hand, we can now prove our non-robust bound, i.e., that Algorithm 1 has loss at most $O\left(\sqrt{b} \cdot \max(EMD(\hat{p}, p), 1)\right)$.

**Theorem 9.** *Let ALG be the corresponding output policy of Algorithm 1. Then we have*

$$c(ALG_p) - c(OPT_p) \leq O\left(\sqrt{b} \cdot \max\left(EMD(\hat{p}, p), 1\right)\right).$$

*Proof.* Since $A_{\hat{K}}$ is the optimal policy for $\hat{p}$, we know from Theorem 8 that

$$c((A_{\hat{K}+\sqrt{b}})_p) - c(OPT_p) \leq O\left(\sqrt{b} \cdot \max\left(EMD(\hat{p}, p), 1\right)\right).$$

Therefore, to prove Theorem 9, we only need to show that

$$c(ALG_p) - c((A_{\hat{K}+\sqrt{b}})_p) \leq O\left(\sqrt{b} \cdot \max\left(EMD(\hat{p}, p), 1\right)\right).$$

We call $c(ALG_p) - c((A_{\hat{K}+\sqrt{b}})_p)$ the expected *truncation loss*. Note that $ALG$ and $A_{\hat{K}+\sqrt{b}}$ differ only if $ALG$ truncates earlier than $A_{\hat{K}+\sqrt{b}}$. This happens when $U + \sqrt{b} < \hat{K} + \sqrt{b}$ and $ALG$ buys on day $U + \sqrt{b} + 1$ instead of $\hat{K} + \sqrt{b} + 1$. In this case, the additional cost incurred by $ALG$ is at most $b$, and this occurs only when the realized rental length satisfies $t > U + \sqrt{b}$. Therefore, the expected truncation loss is bounded by

$$c(ALG_p) - c((A_{\hat{K}+\sqrt{b}})_p) \leq b \cdot \sum_{t>U+\sqrt{b}} p_t.$$

Applying Lemma 7 with $a = U$ and $s = \sqrt{b}$, we obtain

$$\sum_{t>U+\sqrt{b}} p_t \leq \sum_{t>U} \hat{p}_t + \frac{EMD(\hat{p}, p)}{\sqrt{b}}.$$

By the definition of $U$, $\sum_{t>U} \hat{p}_t \leq 1/\sqrt{b}$. Therefore,

$$\sum_{t>U+\sqrt{b}} p_t \leq \frac{1}{\sqrt{b}} + \frac{EMD(\hat{p}, p)}{\sqrt{b}}.$$

Putting these together, we get that

$$c(ALG_p) - c((A_{\hat{K}+\sqrt{b}})_p) \leq b \cdot \left(\frac{1}{\sqrt{b}} + \frac{EMD(\hat{p}, p)}{\sqrt{b}}\right)$$
$$\leq O\left(\sqrt{b} \cdot \max(EMD(\hat{p}, p), 1)\right) \qquad \square$$

Now we want to prove the robustness bound in Theorem 1, i.e., that the additive loss of Algorithm 1 is at most $O(b \log b)$. First let us denote $\hat{Q}_K := \sum_{t>K} \hat{p}_t$, $K = 0, 1, 2, \ldots$. Then a simple calculation implies that

$$c((A_K)_{\hat{p}}) = \sum_{t \leq K} t\hat{p}_t + \sum_{t>K}(K+b)\hat{p}_t = \sum_{i=0}^{K-1} \hat{Q}_i + b\hat{Q}_K \tag{$\triangle$}$$

For $L < \hat{K}$, we have $c((A_{\hat{K}})_{\hat{p}}) \leq c((A_L)_{\hat{p}})$ since $A_{\hat{K}}$ is the optimal policy for $\hat{p}$. Writing this in terms of $\hat{Q}_i$ as above, [2] we get that

$$\sum_{i=0}^{\hat{K}-1} \hat{Q}_i + b\hat{Q}_{\hat{K}} \leq \sum_{i=0}^{L-1} \hat{Q}_i + b\hat{Q}_L$$
$$\implies \sum_{i=L}^{\hat{K}-1} \hat{Q}_i \leq b(\hat{Q}_L - \hat{Q}_{\hat{K}}). \tag{$\S$}$$

We now prove two useful lemmas.

**Lemma 10.** *Let $r = \frac{b-1}{b}$. For every $j = 0, 1, \cdots, \hat{K}$, we have $\hat{Q}_{\hat{K}-j} \geq \frac{\hat{Q}_{\hat{K}}}{r^j}$.*

*Proof.* For simplicity, we write $q := \hat{Q}_{\hat{K}}$. We prove the lemma by induction on $j$.

When $j = 0$, we have $\hat{Q}_{\hat{K}} = q \geq q$, as required.

For the inductive step, assume that $\hat{Q}_{\hat{K}-j} \geq \frac{\hat{Q}_{\hat{K}}}{r^j}$ holds for $1, 2, \cdots, j-1$. Apply ($\S$) with $L = \hat{K} - j$. We have $\sum_{i=\hat{K}-j}^{\hat{K}-1} \hat{Q}_i \leq b(\hat{Q}_{\hat{K}-j} - q)$. Separating the term $\hat{Q}_{\hat{K}-j}$ from the left-hand side and rearranging terms give that

$$(b-1)\hat{Q}_{\hat{K}-j} \geq bq + \sum_{m=1}^{j-1} \hat{Q}_{\hat{K}-m} \geq bq + \sum_{m=1}^{j-1} \frac{q}{r^m}$$

where the last inequality above holds due to the induction hypothesis. Now a standard calculation for a geometric

---

[2]When $A_\infty$ is the optimal policy for $\hat{p}$, we have that for any finite $L$, $\sum_{i \geq L} \hat{Q}_i \leq b\hat{Q}_L$ for all $L$.

series and the fact that $r = \frac{b-1}{b}$ imply that

$$\hat{Q}_{\hat{K}-j} \geq \frac{b}{b-1}q + \frac{1}{b-1}\sum_{m=1}^{j-1}\frac{q}{r^m} = \frac{q}{r^j}. \qquad \square$$

**Lemma 11.** *Let $0 < \beta < \alpha \leq 1$. For any $\gamma \in [0,1]$, define $t(\gamma) := \min\{t \in \mathbb{Z}_{\geq 0} : \hat{Q}_t \leq \gamma\}$. Assume that $t(\alpha) < \hat{K}$ and $t(\beta) \leq \hat{K}$. Then $t(\beta) \leq t(\alpha) + \frac{b\alpha}{\beta}$.*

*Proof.* Since $t(\alpha) < \hat{K}$, taking $L = t(\alpha)$ in inequality (§) gives

$$\sum_{i=t(\alpha)}^{\hat{K}-1} \hat{Q}_i \leq b\,\hat{Q}_{t(\alpha)} \leq b\,\alpha.$$

Since $t(\beta) \leq \hat{K}$, the indices $i = t(\alpha), \ldots, t(\beta) - 1$ are all at most $\hat{K} - 1$. By the definition of $t(\beta)$, we have $\hat{Q}_i > \beta$ for each such $i$. Therefore,

$$\sum_{i=t(\alpha)}^{\hat{K}-1} \hat{Q}_i \geq \sum_{i=t(\alpha)}^{t(\beta)-1} \hat{Q}_i > \big(t(\beta) - t(\alpha)\big)\beta.$$

Combining the two inequalities above yields $\big(t(\beta) - t(\alpha)\big)\beta < b\,\alpha$,[3] which implies the lemma. $\qquad \square$

Now we can prove the second part of Theorem 1.

**Theorem 12.** *Let ALG be the corresponding output policy of Algorithm 1. Then we have*

$$c(ALG_p) - c(OPT_p) \leq O(b\log b).$$

*Proof.* If $\hat{K} < U$, then by the definition of $U$ we have $\hat{Q}_{\hat{K}} > 1/\sqrt{b}$. Taking $j = \hat{K}$ in Lemma 10, we obtain $1 = \hat{Q}_0 \geq \frac{\hat{Q}_{\hat{K}}}{r^{\hat{K}}}$. Equivalently, $\hat{Q}_{\hat{K}} \leq r^{\hat{K}} = \left(\frac{b-1}{b}\right)^{\hat{K}} \leq e^{-\hat{K}/b}$. Combining this with $\hat{Q}_{\hat{K}} > 1/\sqrt{b}$ yields $\frac{1}{\sqrt{b}} < e^{-\hat{K}/b}$, which implies that $\hat{K} < \frac{1}{2}b\log b$. Hence, $K^* \leq O(b\log b)$.

If $U \leq \hat{K}$, we show that $U \leq O(b\log b)$. The idea is to apply Lemma 11 at dyadic tail levels down to $\frac{1}{\sqrt{b}}$. Let $j^* := \lceil \frac{1}{2}\log_2 b \rceil$. Then $2^{-(j^*-1)} > \frac{1}{\sqrt{b}} \geq 2^{-j^*}$. Define $\alpha := 2^{-(j^*-1)}$, so that $\alpha \in (\frac{1}{\sqrt{b}}, \frac{2}{\sqrt{b}}]$.

We first bound $t(\alpha)$. Starting from level 1 and repeatedly halving, we consider the sequence $1, 1/2, 1/4, \ldots, \alpha$. For each intermediate level $\alpha_j := 2^{-j}$ with $j \leq j^* - 1$, we have $\alpha_j > \frac{1}{\sqrt{b}}$, which implies $t(\alpha_j) \leq U \leq \hat{K}$. Therefore, we may iteratively apply Lemma 11 with $\beta = \alpha_j/2$, yielding $t(\alpha) = t(2^{-(j^*-1)}) \leq 2b(j^* - 1)$.

---

[3] When the optimal policy for $\hat{p}$ is $A_\infty$, we are able to follow the same route and get $t(\frac{\alpha}{2}) < t(\alpha) + 2b$, for every $\alpha \in (0,1]$.

We now consider the final step from level $\alpha$ down to $\frac{1}{\sqrt{b}}$. Applying Lemma 11 with $\alpha = 2^{-(j^*-1)}$ and $\beta = \frac{1}{\sqrt{b}}$, and using the fact that $t(\beta) = U \leq \hat{K}$, we obtain

$$U = t(1/\sqrt{b}) \leq t(\alpha) + \frac{b\alpha}{1/\sqrt{b}} = t(\alpha) + b\alpha\sqrt{b}.$$

Since $\alpha \leq \frac{2}{\sqrt{b}}$, we have $b\alpha\sqrt{b} \leq 2b$, and hence $U \leq t(\alpha) + 2b$. Combining with the bound on $t(\alpha)$ gives $U \leq 2b(j^*-1)+2b = 2bj^* = 2b\lceil\frac{1}{2}\log_2 b\rceil = O(b\log b)$. Thus, when $U \leq \hat{K}$, we have $U \leq O(b\log b)$.[4]

This implies that Algorithm 1 always truncates by time $O(b\log b)$, and therefore incurs cost (and additive loss) at most $O(b\log b)$. $\qquad \square$

*Proof of Theorem 1.* This is directly implied by taking the minimum of the two bounds in Theorems 9 and 12. $\qquad \square$

## 4. Consistency vs. Robustness Tradeoff

In this section, we show that the robustness–consistency tradeoff achieved by our main algorithm is pareto-optimal (Theorem 6): we prove that when the prediction error is unknown, any algorithm with consistency $O(\sqrt{b})$ (the additive loss when the prediction error is 0) must have robustness $\Omega(b\log b)$ (additive loss when the prediction error is arbitrarily large). Combined with the upper bound in Theorem 1, this implies that the robustness–consistency tradeoff of Algorithm 1 is tight.

**Theorem 6.** *For any deterministic or randomized algorithm ALG, if $c(ALG_p) - c(OPT_p) \leq O\big(\sqrt{b}\big)$ when $EMD(\hat{p}, p) = 0$, then $c(ALG_p) - c(OPT_p) \geq \Omega(b\log b)$ for large enough prediction error $EMD(\hat{p}, p)$.*

*Proof.* Let $r := 1 - \frac{2}{b}$ and choose $K := \lceil\frac{1}{2}b\log b\rceil$. We first construct a prediction $\hat{p}$.

We use the same notation $\hat{Q}$ to represent the tail probability mass in $\hat{p}$ as we did in Section 3. Let $\hat{Q}_0 = 1$, and for $t = 0, 1, \ldots, K-1$, define $\hat{Q}_{t+1} := r\hat{Q}_t$, so that $\hat{Q}_t = r^t$ up to time $K$. For $t \geq K$, we switch to a slower decay and define $\hat{Q}_{t+1} := (1 - \frac{1}{2b})\hat{Q}_t$. We then define $\hat{p}$ by setting $\hat{p}_{t+1} := \hat{Q}_t - \hat{Q}_{t+1}$. By construction, $\sum_t \hat{p}_t = 1$. Since our setting assumes distributions with finite support $[N]$, one may replace $\hat{p}$ by a finite-support distribution obtained by aggregating all probability mass beyond $N$ and assigning it to $N$, for a sufficiently large $N$. This preserves $\sum_t \hat{p}_t = 1$

---

[4] When $A_\infty$ is the optimal policy for $\hat{p}$, we do not need to separate the final step because the lemma holds for all $\alpha$. In this case, just iterating the lemma starting from $\alpha = 1$ and after $m$ halvings we have $t(2^{-m}) \leq 2bm$. The result holds by choosing $m$ with $2^{-m} \leq \frac{1}{\sqrt{b}}$.

and does not affect the analysis; for simplicity, we continue to denote the resulting distribution by $\hat{p}$.

We now compute the optimal policy for $\hat{p}$. Equation ($\triangle$) implies that $c((A_{t+1})_{\hat{p}}) - c((A_t)_{\hat{p}}) = \hat{Q}_t - b\hat{p}_{t+1}$. For $t < K$, we have $\hat{Q}_{t+1} = r\hat{Q}_t$, and hence $\hat{p}_{t+1} = \hat{Q}_t - \hat{Q}_{t+1} = (1-r)\hat{Q}_t = \frac{2}{b}\hat{Q}_t$. It follows that $c((A_{t+1})_{\hat{p}}) - c((A_t)_{\hat{p}}) = \hat{Q}_t - 2\hat{Q}_t = -\hat{Q}_t < 0$. Therefore, $c((A_{t+1})_{\hat{p}}) < c((A_t)_{\hat{p}})$, and the cost strictly decreases up to time $K$.

For $t \geq K$, we have $\hat{p}_{t+1} = \frac{1}{2b}\hat{Q}_t$, which implies that $c((A_{t+1})_{\hat{p}}) - c((A_t)_{\hat{p}}) = \hat{Q}_t - \frac{b}{2b}\hat{Q}_t = \frac{1}{2}\hat{Q}_t > 0$. Thus, $c((A_{t+1})_{\hat{p}}) > c((A_t)_{\hat{p}})$, and the cost strictly increases for all $t \geq K$. Since the cost increases for all $t \geq K$, the limit $A_\infty$ is strictly worse than $A_K$ as well. Consequently, $A_K$ is the optimal policy for $\hat{p}$.

We first prove the lower bound for deterministic algorithms, and then extend it to randomized algorithms.

Any deterministic algorithm corresponds to a policy $A_B$ for some fixed time $B \in \{0, 1, 2, \ldots, \infty\}$. Assume that $A_B$ satisfies the consistency guarantee, i.e., assume that $c((A_B)_p) - c((A_K)_p) \leq O(\sqrt{b})$ when $p = \hat{p}$. We first show that this implies that $B = \Omega(b \log b)$.

Since $K = \Omega(b \log b)$, if $B \geq K$ then we immediately have $B = \Omega(b \log b)$. So we may assume that $B < K$.

Recall that equation ($\triangle$) says that $c((A_B)_p) = \sum_{i=0}^{B-1} \hat{Q}_i + b\hat{Q}_B$. For $i \leq K$, we have $\hat{Q}_i = r^i$, and hence $c((A_B)_p) = \frac{b}{2}(1 - r^B) + br^B = \frac{b}{2} + \frac{b}{2}r^B$ and $c((A_K)_p) = \frac{b}{2} + \frac{b}{2}r^K$. Therefore, the consistency guarantee implies that $\frac{b}{2}(r^B - r^K) \leq C\sqrt{b}$ for some constant $C$, or equivalently, $r^B \leq r^K + \frac{2C}{\sqrt{b}}$. Since $r^K = (1 - \frac{2}{b})^K \leq e^{-2K/b} \leq e^{-\log b} = 1/b \leq C/\sqrt{b}$ for large $b$, we obtain $r^B \leq \frac{3C}{\sqrt{b}}$. Using the inequality $\log(1 - x) \geq -x/(1-x)$ for $0 < x < 1$ with $x = 2/b$, we have $\log r = \log(1 - \frac{2}{b}) \geq -\frac{2}{b-2}$. Thus $r^B = e^{B \log r} \geq e^{-2B/(b-2)}$. Combining the above bounds yields $e^{-2B/(b-2)} \leq \frac{3C}{\sqrt{b}}$, which implies $\frac{2B}{b-2} \geq \frac{1}{2}\log b - \log(3C)$. Therefore, $B \geq \frac{b-2}{4}\log b - \frac{b-2}{2}\log(3C) = \Omega(b \log b)$.

So we have shown that any deterministic algorithm $A_B$ satisfying the consistency guarantee must satisfy $B = \Omega(b \log b)$. Suppose the true distribution is $p$ with $p_{b^{100}} = 1$ and $p_t = 0$ for all $t \neq b^{100}$. In this case, the optimal policy is to pay $b$ to buy at the beginning, while $A_B$ will pay $\Omega(b \log b)$ to rent until time $B$. So the additive loss is $\Omega(b \log b)$ as claimed.

A randomized algorithm can be viewed as a convex combination of deterministic algorithms. Equivalently, it samples a threshold $B$ from some distribution and then runs the policy $A_B$. Define $\Delta(B) := c((A_B)_p) - c((A_K)_p)$. Thus, a randomized algorithm satisfying the consistency guarantee must satisfy $\mathbb{E}[\Delta(B)] \leq C\sqrt{b}$ for some constant $C$. Note that the function $B \mapsto \Delta(B)$ is strictly decreasing on

$B \leq K$ since $r^B$ is strictly decreasing. Let $B_0$ be the largest value such that $\Delta(B_0) \geq 2C\sqrt{b}$. Then for all $B \leq B_0$, we have $\Delta(B) \geq 2C\sqrt{b}$. Since $\Delta(B) \geq 0$ for all $B$, it follows that $\Pr(B \leq B_0) \cdot 2C\sqrt{b} \leq \mathbb{E}[\Delta(B)] \leq C\sqrt{b}$, and hence $\Pr(B \leq B_0) \leq \frac{1}{2}$. Therefore, $\mathbb{E}[B] \geq \Pr(B > B_0) \cdot B_0 \geq \frac{1}{2}B_0$. On the other hand, since $\Delta(B_0) \geq 2C\sqrt{b}$ and $\Delta(B_0 + 1) < 2C\sqrt{b}$, following the same reasoning as in the deterministic case implies that $r^{B_0} = \Theta(\frac{1}{\sqrt{b}})$. Taking logarithms yields $B_0 = \Theta(b \log b)$. Consequently, $\mathbb{E}[B] \geq \Omega(b \log b)$. Finally, letting the true distribution be $p$ with $p_{b^{100}} = 1$ and $p_t = 0$ for all $t \neq b^{100}$ shows that the additive loss is $\Omega(b \log b)$. This completes the proof for randomized algorithms. $\square$

## Acknowledgements

This work was supported in part by NSF award 2228995.

## Impact Statement

This paper presents work whose goal is to advance the field of Machine Learning. As a theoretical result in the well-established field of online algorithms and the newer field of algorithms with predictions, we believe the main impact of our work is to advance human knowledge. While there may be potential societal consequences of our work, we do not feel that any must be specifically highlighted here.

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

## A. Further Robustification

When $EMD(\hat{p}, p)$ is large (larger than $\sqrt{b}\log b$), Algorithm 1 could incur a $\Theta(b\log b)$ cost, which is a robustness guarantee by the algorithm itself for free. However, it is still desirable to explicitly interpolate between this algorithm and the classical worst-case strategy to achieve a stronger robustness guarantee. In this section, we present a further robustification of Algorithm 1 that preserves benefits of the prediction and the free robustness while providing additional explicit worst-case control.

Recall that the classical worst-case algorithm for the ski rental problem buys skis at the beginning of day $b$, achieving the optimal competitive ratio of 2 and additive loss of $b$ (Karlin et al., 1986). A natural idea to robustify our algorithm is to "push" our algorithm closer to the traditional worst-case algorithm, making them as similar as possible. To do this, we introduce a hyperparameter $\lambda \in [0, 1]$ that controls the degree of robustification. Algorithm 2 is a robustification of Algorithm 1 that uses $\lambda$ to decide how much to "push" Algorithm 1 towards the worst-case algorithm.

As before, we assume without loss of generality that $\sqrt{b}$ is an integer; otherwise, we use $\lfloor \sqrt{b} \rfloor$ instead.

---

**Algorithm 2** Further robustification of Algorithm 1

---

1: Compute the optimal policy $A_{\hat{K}}$ for the prediction $\hat{p}$.
2: Compute $U$ such that $\sum_{t>U} \hat{p}_t \leq \frac{1}{\sqrt{b}}$.
3: Set $K^* = \min(\hat{K} + \sqrt{b}, U + \sqrt{b})$.
4: **if** $K^* \leq b$ **then**
5:    **if** $K^* < \lceil \lambda b \rceil$ **then**
6:       Buy at the beginning of day $\lceil \lambda b \rceil$.
7:    **else**
8:       Buy at the beginning of day $K^* + 1$.
9:    **end if**
10: **else**
11:    **if** $K^* \geq \lceil b/\lambda \rceil$ **then**
12:       Buy at the beginning of day $\lceil b/\lambda \rceil$.
13:    **else**
14:       Buy at the beginning of day $K^* + 1$.
15:    **end if**
16: **end if**

---

We have the following theorem.

**Theorem 2.** *There is an algorithm ALG which takes as input a parameter $\lambda \in [0, 1]$ and a distributional prediction $\hat{p}$, and has expected cost*

$$c(ALG_p) \leq \min\Big\{(1+\lambda)\big(c(OPT_p) + \mathcal{B}_{\hat{p},p}\big),$$
$$\Big(1 + \frac{1}{\lambda}\Big)c(OPT_p)\Big\},$$

*where* $\mathcal{B}_{\hat{p},p} = O\left(\min\left(\sqrt{b}\max\left(EMD(\hat{p},p),1\right), b\log b\right)\right)$.

*Proof of Theorem 2.* If $\lambda = 0$, we run Algorithm 1 so that the first term obviously holds. The second term goes to $\infty$, thus Algorithm 1 is the required algorithm. Now, let us consider the main case when $\lambda > 0$ where we use Algorithm 2. In the following analysis, we break into cases depending on the output of Algorithm 2.

**Case 1.** If the output of Algorithm 2 is to buy at the beginning of day $\lceil \lambda b \rceil$. We know that in this case we have $K^* < \lceil \lambda b \rceil$.

First, let us show the second part of the bound (which leads to the robustness), i.e., we want to show that $c((A_{\lceil \lambda b \rceil - 1})_p) \leq (1 + \frac{1}{\lambda})c(OPT_p)$. Here $LHS = \sum_{t \leq \lceil \lambda b \rceil - 1} tp_t + \sum_{t \geq \lceil \lambda b \rceil}(\lceil \lambda b \rceil - 1 + b)p_t$.

If $OPT_p = A_\infty$, then $RHS = (1 + \frac{1}{\lambda})\sum_t tp_t$. To show that $LHS \leq RHS$, we only need to show that $\sum_{t \geq \lceil \lambda b \rceil}(\lceil \lambda b \rceil - 1 + b)p_t \leq (1 + \frac{1}{\lambda})\sum_{t \geq \lceil \lambda b \rceil} tp_t$. This

holds because

$$\sum_{t \geq \lceil \lambda b \rceil}(\lceil \lambda b \rceil - 1 + b)p_t \leq \sum_{t \geq \lceil \lambda b \rceil}(\lambda b + b)p_t$$
$$= \frac{\lambda + 1}{\lambda}\sum_{t \geq \lceil \lambda b \rceil}(\lambda b)p_t$$
$$\leq \Big(1 + \frac{1}{\lambda}\Big)\sum_{t \geq \lceil \lambda b \rceil} tp_t.$$

If $OPT_p = A_K$, then $RHS = (1 + \frac{1}{\lambda})\left[\sum_{t \leq K} tp_t + \sum_{t > K}(K+b)p_t\right]$. We check that the coefficient of term $p_t$ in the $LHS$ is at most that in the $RHS$ for each $t$. If $K \leq \lceil \lambda b \rceil - 1$, then:

For $t \leq K : t \leq \Big(1 + \frac{1}{\lambda}\Big)t.$

For $K < t \leq \lceil \lambda b \rceil - 1 : t \leq K + b \leq (1 + \frac{1}{\lambda})(K + b).$

For $t \geq \lceil \lambda b \rceil : \lceil \lambda b \rceil - 1 + b \leq \Big(1 + \frac{1}{\lambda}\Big)\lceil \lambda b \rceil$
$$\leq \Big(1 + \frac{1}{\lambda}\Big)(K + b).$$

If $\lceil \lambda b \rceil - 1 < K$, then:

For $t \leq \lceil \lambda b \rceil - 1 : t \leq \Big(1 + \frac{1}{\lambda}\Big)t.$

For $\lceil \lambda b \rceil \leq t \leq K : \lceil \lambda b \rceil - 1 + b \leq (1 + \frac{1}{\lambda})\lceil \lambda b \rceil$
$$\leq (1 + \frac{1}{\lambda})t.$$

For $t > K : \lceil \lambda b \rceil - 1 + b \leq \Big(1 + \frac{1}{\lambda}\Big)\lceil \lambda b \rceil$
$$\leq \Big(1 + \frac{1}{\lambda}\Big)(K + b).$$

Then, let us show the first part of the bound (which leads to the consistency).

We know from Theorem 1 that $c((A_{K^*})_p) \leq c(OPT_p) + \min\left(O(\sqrt{b}\max(EMD(\hat{p},p),1)), O(b\log b)\right)$. Thus, we only need to show that $c((A_{\lceil \lambda b \rceil - 1})_p) \leq (1+\lambda)c((A_{K^*})_p)$. That is, we want to show

$$\sum_{t \leq \lceil \lambda b \rceil - 1} tp_t + \sum_{t \geq \lceil \lambda b \rceil}(\lceil \lambda b \rceil - 1 + b)p_t$$
$$\leq (1 + \lambda)\left[\sum_{t \leq K^*} tp_t + \sum_{t > K^*}(K^* + b)p_t\right].$$

Note that $K^* \leq \lceil \lambda b \rceil - 1$, the above inequality holds since

For $t \leq K^* : t \leq (1 + \lambda)t.$
For $K^* < t \leq \lceil \lambda b \rceil - 1 : t \leq \lceil \lambda b \rceil - 1 < (1 + \lambda)(K^* + b).$
For $t \geq \lceil \lambda b \rceil : \lceil \lambda b \rceil - 1 + b < \lambda b + b \leq (1 + \lambda)(K^* + b).$

**Case 2.** If the output of Algorithm 2 is to buy at the beginning of day $\lceil b/\lambda \rceil$. We know that in this case we have $K^* \geq \lceil b/\lambda \rceil$.

First, let us show the second part of the bound. We want to show that $c((A_{\lceil b/\lambda \rceil - 1})_p) \leq (1 + \frac{1}{\lambda})c(OPT_p)$. Here, $LHS = \sum_{t \leq \lceil b/\lambda \rceil - 1} tp_t + \sum_{t \geq \lceil b/\lambda \rceil}(\lceil b/\lambda \rceil - 1 + b)p_t$.

If $OPT_p = A_\infty$, then $RHS = (1 + \frac{1}{\lambda})\sum_t tp_t$. To show that $LHS \leq RHS$, we only need to show that $\sum_{t \geq \lceil b/\lambda \rceil}(\lceil b/\lambda \rceil - 1 + b)p_t \leq (1 + \frac{1}{\lambda})\sum_{t \geq \lceil b/\lambda \rceil} tp_t$. This holds because

$$\sum_{t \geq \lceil \frac{b}{\lambda} \rceil}(\lceil \tfrac{b}{\lambda} \rceil - 1 + b)p_t \leq \sum_{t \geq \lceil \frac{b}{\lambda} \rceil}\left(\lceil \tfrac{b}{\lambda} \rceil + \tfrac{1}{\lambda}\lceil \tfrac{b}{\lambda} \rceil\right)p_t$$

$$= \frac{\lambda + 1}{\lambda}\sum_{t \geq \lceil \frac{b}{\lambda} \rceil}\lceil \tfrac{b}{\lambda} \rceil p_t$$

$$\leq \left(1 + \frac{1}{\lambda}\right)\sum_{t \geq \lceil \frac{b}{\lambda} \rceil} tp_t.$$

If $OPT_p = A_K$, then $RHS = (1 + \frac{1}{\lambda})\left[\sum_{t \leq K} tp_t + \sum_{t > K}(K + b)p_t\right]$. We check that the coefficient of term $p_t$ in the $LHS$ is at most that in the $RHS$ for each $t$. If $K \leq \lceil \frac{b}{\lambda} \rceil - 1$, then:

For $t \leq K$: $t \leq \left(1 + \frac{1}{\lambda}\right)t$.

For $K < t \leq \lceil \frac{b}{\lambda} \rceil - 1$: $t \leq \lceil \frac{b}{\lambda} \rceil - 1 < \left(1 + \frac{1}{\lambda}\right)(K + b)$.

For $t \geq \lceil \frac{b}{\lambda} \rceil$: $\lceil \frac{b}{\lambda} \rceil - 1 + b \leq \frac{b}{\lambda} + b \leq \left(1 + \frac{1}{\lambda}\right)(K + b)$.

If $\lceil \frac{b}{\lambda} \rceil - 1 < K$, then:

For $t \leq \lceil \frac{b}{\lambda} \rceil - 1$: $t \leq \left(1 + \frac{1}{\lambda}\right)t$.

For $\lceil \frac{b}{\lambda} \rceil - 1 < t \leq K$: $\lceil \frac{b}{\lambda} \rceil - 1 + b \leq \left(1 + \frac{1}{\lambda}\right)(\lceil \frac{b}{\lambda} \rceil - 1)$.

For $t > K$: $\lceil \frac{b}{\lambda} \rceil - 1 + b \leq \frac{b}{\lambda} + b \leq \left(1 + \frac{1}{\lambda}\right)(K + b)$.

Then, let us show the first part of the bound.

We know from Theorem 1 that $c((A_{K^*})_p) \leq c(OPT_p) + \min\left(O(\sqrt{b}\max(EMD(\hat{p}, p), 1)), O(b\log b)\right)$. Thus, we only need to show that $c((A_{\lceil b/\lambda \rceil - 1})_p) \leq (1 + \lambda)c((A_{K^*})_p)$. That is, we want to show

$$\sum_{t \leq \lceil \frac{b}{\lambda} \rceil - 1} tp_t + \sum_{t \geq \lceil \frac{b}{\lambda} \rceil}(\lceil \tfrac{b}{\lambda} \rceil - 1 + b)p_t$$

$$\leq (1 + \lambda)\left[\sum_{t \leq K^*} tp_t + \sum_{t > K^*}(K^* + b)p_t\right].$$

Note that $\lceil b/\lambda \rceil \leq K^*$, the above inequality holds since

For $t \leq \lceil \frac{b}{\lambda} \rceil - 1$: $t \leq (1 + \lambda)t$.

For $\lceil \frac{b}{\lambda} \rceil \leq t \leq K^*$: $\lceil \frac{b}{\lambda} \rceil - 1 + b \leq \lceil \frac{b}{\lambda} \rceil + \lambda\lceil \frac{b}{\lambda} \rceil$

$$\leq (1 + \lambda)t.$$

For $t > K^*$: $\lceil \frac{b}{\lambda} \rceil - 1 + b \leq (1 + \lambda)\lceil \frac{b}{\lambda} \rceil$

$$\leq (1 + \lambda)(K^* + b).$$

**Case 3.** If the output of Algorithm 2 is to buy at the beginning of day $K^* + 1$. We know that in this case we have $\lceil \lambda b \rceil \leq K^* < \lceil b/\lambda \rceil$.

For the first part of the bound, we know from Theorem 1 that $c((A_{K^*})_p) \leq c(OPT_p) + \min\left(O(\sqrt{b}\max(EMD(\hat{p}, p), 1)), O(b\log b)\right)$. The cost of our algorithm is exactly $c((A_{K^*})_p)$, thus it holds trivially.

For the second part of the bound, we want to show that $c((A_{K^*})_p) \leq (1 + \frac{1}{\lambda})c(OPT_p)$. Here, $LHS = \sum_{t \leq K^*} tp_t + \sum_{t > K^*}(K^* + b)p_t$.

If $OPT_p = A_\infty$, then $RHS = (1 + \frac{1}{\lambda})\sum_t tp_t$. To show that $LHS \leq RHS$, we only need to show that $\sum_{t > K^*}(K^* + b)p_t \leq (1 + \frac{1}{\lambda})\sum_{t > K^*} tp_t$. This holds because

$$\sum_{t > K^*}(K^* + b)p_t \leq \sum_{t > K^*}\left(t + \tfrac{1}{\lambda}\lceil \lambda b \rceil\right)p_t$$

$$\leq \sum_{t > K^*}\left(t + \tfrac{1}{\lambda}K^*\right)p_t$$

$$\leq \sum_{t > K^*}\left(1 + \tfrac{1}{\lambda}\right)t\, p_t.$$

If $OPT_p = A_K$, then $RHS = (1 + \frac{1}{\lambda})\left[\sum_{t \leq K} tp_t + \sum_{t > K}(K + b)p_t\right]$. We check that the coefficient of term $p_t$ in the $LHS$ is at most that in the $RHS$ for each $t$. If $K \leq K^*$, then:

For $t \leq K$: $t \leq \left(1 + \frac{1}{\lambda}\right)t$.

For $K < t \leq K^*$: $t \leq K^* \leq \lceil \frac{b}{\lambda} \rceil - 1 < \left(1 + \frac{1}{\lambda}\right)(K + b)$.

For $t > K^*$: $K^* + b < \frac{b}{\lambda} + b \leq \left(1 + \frac{1}{\lambda}\right)(K + b)$.

If $K^* < K$, then:

For $t \leq K^*$ : $t \leq \left(1 + \frac{1}{\lambda}\right)t$.

For $K^* < t \leq K$ : $K^* + b \leq t + \frac{1}{\lambda}\lceil \lambda b \rceil \leq t + \frac{1}{\lambda}K^*$

$$\leq \left(1 + \frac{1}{\lambda}\right)t.$$

For $t > K$ : $K^* + b < \frac{b}{\lambda} + b = \left(1 + \frac{1}{\lambda}\right)b$

$$\leq \left(1 + \frac{1}{\lambda}\right)(K + b).$$

Moreover,

$$\sum_{t > K^*} (K^* + b)p_t \leq \sum_{t > K^*} \left(t + \frac{1}{\lambda}\lceil \lambda b \rceil\right)p_t$$

$$\leq \sum_{t > K^*} \left(t + \frac{1}{\lambda}K^*\right)p_t$$

$$\leq \sum_{t > K^*} \left(1 + \frac{1}{\lambda}\right)t\, p_t.$$

Thus, further robustness is proven as required. $\square$

We get the following corollary of Theorem 2 by just dividing both sides by $OPT_p$; this form is directly analogous to the bounds of Purohit et al. (2018).

**Corollary 13.** *There is an algorithm ALG which takes as input a parameter* $\lambda \in [0, 1]$ *and a distributional prediction* $\hat{p}$, *and has*

$$\frac{c(ALG_p)}{c(OPT_p)} \leq \min\left\{(1 + \lambda)\left(1 + \frac{\mathcal{B}_{\hat{p},p}}{c(OPT_p)}\right), 1 + \frac{1}{\lambda}\right\}.$$

*where* $\mathcal{B}_{\hat{p},p} = O\left(\min\left(\sqrt{b}\max\left(EMD(\hat{p}, p), 1\right),\, b\log b\right)\right)$.

## B. Technical Proofs of Lower Bounds

This section gives the technical proofs of the three lower bounds (Theorems 3 4 5) mentioned in Section 2.2.

### Proof of Theorem 3

*Proof.* Let $r = 1 - \epsilon$. We assume that $b^{\frac{r+1}{2}}$ is an integer; otherwise, we consider $\lfloor b^{\frac{r+1}{2}} \rfloor$ instead. Consider the following $b^{\frac{r+1}{2}} + 2$ different probability distributions $p^{(j)}, 1 \leq j \leq b^{\frac{r+1}{2}} + 2$, where

$$p_t^{(j)} = \begin{cases} \delta & \text{if } t = j \\ 1 - \delta & \text{if } t = b^{\frac{r+1}{2}} + 1 + b \end{cases}$$

and $\delta = b^{\frac{r+1}{2} - 1} + \frac{2}{b} + \frac{1}{b^2}$. Note that the $EMD$ between any two distributions is bounded by $O(b^{\frac{r+1}{2}} \cdot b^{\frac{r+1}{2} - 1}) = O(b^r)$.

Consider any prediction $\hat{p}$ such that $EMD(\hat{p}, p^{(j)}) \leq O(b^r)$ for every $1 \leq j \leq b^{\frac{r+1}{2}} + 2$. We claim that when $\hat{p}$ is our prediction, there is no algorithm $ALG$ such that the additive loss $\text{diff}_p$ is bounded by $O(EMD(\hat{p}, p)) + O(b^r)$.

To show this, let $ALG$ be the algorithm we use. If $ALG = A_K$ for some $0 \leq K \leq b^{\frac{r+1}{2}}$, consider that $p^{(K+1)}$ is the truth. In this case, $c(ALG_p) = K + b$. Due to the choice of $\delta$, we know that $A_{K+1}$ is the optimal policy for $p^{(K+1)}$. Thus, the optimal policy has cost $c(OPT_p) = (K+1)\delta + (K+1+b)(1-\delta)$. Then, we have $\text{diff}_p = b\delta - 1 \geq \Omega(b^{\frac{r+1}{2}})$. Based on the fact that $\frac{r+1}{2} > r$, this proves the claim when $K \leq b^{\frac{r+1}{2}}$.

If $ALG = A_K$ for some $K \geq b^{\frac{r+1}{2}} + 1$ or $K = \infty$, consider that $p^{(1)}$ is the truth. Then by the construction of $p^{(1)}$, we see that $c((A_K)_p) \geq c((A_\infty)_p)$ for every $K \geq b^{\frac{r+1}{2}} + 1$. In fact, the algorithms $A_K$ can be trivially ruled out for finite $K \geq b^{\frac{r+1}{2}} + 1$ since the algorithm has to be bad even with the promise that the truth is from one of the $p^{(j)}$ inputs. Thus, it suffices to prove the claim for $ALG = A_\infty$. In this case, $c((A_\infty)_p) = 1 \cdot \delta + (b^{\frac{r+1}{2}} + 1 + b)(1 - \delta)$ and $c(OPT_p) = 1 \cdot \delta + (b+1)(1-\delta)$. Then, we have $\text{diff}_p = b^{\frac{r+1}{2}}(1 - \delta) \geq \Omega(b^{\frac{r+1}{2}})$. This proves the theorem for all deterministic algorithms.

To show that this theorem holds even for randomized algorithms, we use Yao's Lemma to make our algorithm $ALG = A_K$ deterministically for some $K$ and choose the truth uniformly at random from $p^{(j)}, 1 \leq j \leq b^{\frac{r+1}{2}} + 2$. When $K \leq b^{\frac{r+1}{2}}/2$, there is a constant probability that we choose the truth $p^{(j)}$ where $j$ lies in the next $\Omega(b^{\frac{r+1}{2}})$ interval (e.g. $b^{\frac{r+1}{2}}/4$) passed $K$ because of the uniformly chosen. For any truth $p^{(T)}$ where $T = K + x$ lies in the interval, we have $c((A_K)_{p^{(T)}}) = K + b$ and $c(OPT_{p^{(T)}}) = (K+x)\delta + (K+x+b)(1-\delta)$. Thus, $\text{diff}_p = b\delta - x \geq \Omega(b^{\frac{r+1}{2}})$. When $K \geq b^{\frac{r+1}{2}}/2$, there is a constant probability that we choose the truth $p^{(j)}$ where $j$ lies in the $\Omega(b^{\frac{r+1}{2}})$ prefix (e.g. $[1, b^{\frac{r+1}{2}}/4]$). Again, for any truth $p^{(T)}$ where $T = x$ lies in the above prefix, we have $c((A_K)_{p^{(T)}}) = x\delta + (K+b)(1-\delta)$ and $c(OPT_{p^{(T)}}) = x\delta + (x+b)(1-\delta)$. Thus, $\text{diff}_p = (K - x)(1 - \delta) \geq \Omega(b^{\frac{r+1}{2}})$. It turns out that we also have this lower bound against randomized algorithms, meaning that even for a randomized algorithm, its expected cost cannot be within the additive loss $O(EMD(\hat{p}, p)) + O(b^{1-\epsilon})$ of the expected cost to the optimum. $\square$

**Proof of Theorem 4**

*Proof.* Define two probability distributions

$$p_t^1 = \begin{cases} \frac{1}{2} & \text{if } t = 1, \\ \frac{1}{4} & \text{if } t = 2, \\ \epsilon = \frac{1}{2b} + \delta & \text{if } t = 4, \\ \frac{1}{4} - \epsilon & \text{if } t = b + 3, \end{cases}$$

and

$$p_t^2 = \begin{cases} \frac{1}{2} & \text{if } t = 1, \\ \frac{1}{4} + \epsilon & \text{if } t = 2, \\ 0 & \text{if } t = 4, \\ \frac{1}{4} - \epsilon & \text{if } t = b + 3, \end{cases}$$

where $\delta > 0$. One can check that $A_\infty$ is the optimal policy for $p^1$ and $A_2$ is the optimal policy for $p^2$. Notice that $EMD(p^1, p^2) = 2(\frac{1}{2b} + \delta) = \frac{1}{b} + 2\delta \leq O(\frac{1}{b})$ for sufficiently small $\delta$. Now, assume that we receive some predicted distribution $\hat{p}$ such that $EMD(\hat{p}, p^1) \leq O(\frac{1}{b})$ and $EMD(\hat{p}, p^2) \leq O(\frac{1}{b})$. We claim that given the information of this $\hat{p}$, there is no $ALG$ such that we have both

$$\text{diff}_1 = c(ALG_{p^1}) - c(OPT_{p^1}) \leq o(b) \cdot EMD(\hat{p}, p^1)$$

and

$$\text{diff}_2 = c(ALG_{p^2}) - c(OPT_{p^2}) \leq o(b) \cdot EMD(\hat{p}, p^2).$$

Note that if this claim holds, then it is impossible to find an algorithm such that the additive loss is bounded by $o(b) \cdot EMD(\hat{p}, p)$. This is because, based on the information of $\hat{p}$, for any algorithm $ALG$ we choose, at least one of the two formulas above must fail to hold, and the corresponding $p^i$ in the failed formula could possibly be the unknown true distribution $p$ of the problem.

To show this claim, let us use $ALG = A_K$. When $0 \leq K \leq 3$, consider that $p^1$ is the truth. Then, one can check that for each $K = 0, 1, 2, 3$, $\text{diff}_1 \geq \Omega(1)$. By the fact that $EMD(\hat{p}, p^1) \leq O(\frac{1}{b})$, we have $\text{diff}_1 \geq \Omega(b) \cdot EMD(\hat{p}, p^1)$. Thus, the first formula fails. When $K \geq 4$ or $K = \infty$, consider that $p^2$ is the truth. Then, one can check that for each $K \geq 4$ or $K = \infty$, $\text{diff}_2 \geq \Omega(1)$. Hence, we have $\text{diff}_2 \geq \Omega(b) \cdot EMD(\hat{p}, p^2)$. This concludes that for any $ALG$ (deterministic or randomized[5]) we choose, there exists some truth $p^i \in \{p^1, p^2\}$ such that using $ALG$ we have $\text{diff}_i \geq \Omega(b) \cdot EMD(\hat{p}, p^i)$. $\square$

**Proof of Theorem 5**

*Proof.* Consider two distributions $p^1$ and $p^2$ defined as follows.

---

[5]Choose the truth $p^i$ such that the total probability mass of deterministic algorithms that perform poorly under $p^i$ is bounded below by a constant, thus holds for any randomized algorithm by Yao's Lemma.

$$p_t^1 = \begin{cases} \frac{1}{2} & \text{if } t = 1, \\ \epsilon = \frac{1}{\sqrt{b}} & \text{if } t = \sqrt{b}, \\ \frac{1}{2} - \epsilon & \text{if } t = b - 1 + \sqrt{b}, \end{cases}$$

and

$$p_t^2 = \begin{cases} \frac{1}{2} + \epsilon & \text{if } t = 1, \\ 0 & \text{if } t = \sqrt{b}, \\ \frac{1}{2} - \epsilon & \text{if } t = b - 1 + \sqrt{b}. \end{cases}$$

One can check that $A_\infty$ is the optimal policy for $p^1$ and $A_1$ is the optimal policy for $p^2$. Notice that $EMD(p^1, p^2) = \epsilon \cdot (\sqrt{b} - 1) = \Theta(1)$. Now, assume that we receive some predicted distribution $\hat{p}$ such that $EMD(\hat{p}, p^1) = \Theta(1)$ and $EMD(\hat{p}, p^2) = \Theta(1)$. We claim that given the information of this $\hat{p}$, there is no $ALG$ such that we have both

$$\text{diff}_1 = c(ALG_{p^1}) - c(OPT_{p^1}) \leq o(\sqrt{b}) \cdot \max(EMD(\hat{p}, p), 1)$$

and

$$\text{diff}_2 = c(ALG_{p^2}) - c(OPT_{p^2}) \leq o(\sqrt{b}) \cdot \max(EMD(\hat{p}, p), 1).$$

Note that if this claim holds, then it is impossible to find an algorithm such that the additive loss is bounded by $o(\sqrt{b}) \cdot \max(EMD(\hat{p}, p), 1)$. This is because, based on the information of $\hat{p}$, for any algorithm $ALG$ we choose, at least one of the two formulas above must fail to hold, and the corresponding $p^i$ in the failed formula could possibly be the unknown true distribution $p$ of the problem.

To show this claim, let us use $ALG = A_K$. When $0 \leq K \leq \sqrt{b} - 1$, consider that $p^1$ is the truth. Then, one can check that for each $K \leq \sqrt{b} - 1$, $\text{diff}_1 \geq \Omega(\sqrt{b})$. By the fact that $EMD(\hat{p}, p^1) = \Theta(1)$, we have $\text{diff}_1 \geq \Omega(\sqrt{b}) \cdot \max(EMD(\hat{p}, p), 1)$. Thus, the first formula fails. When $K \geq \sqrt{b}$ or $K = \infty$, consider that $p^2$ is the truth. Then, one can check that for each $K \geq \sqrt{b}$ or $K = \infty$, $\text{diff}_2 \geq \Omega(\sqrt{b})$. Hence, we have $\text{diff}_2 \geq \Omega(\sqrt{b}) \cdot \max(EMD(\hat{p}, p), 1)$. This concludes that for any $ALG$ (deterministic or randomized[6]) we choose, there exists some truth $p^i \in \{p^1, p^2\}$ such that using $ALG$ we have $\text{diff}_i \geq \Omega(\sqrt{b}) \cdot \max(EMD(\hat{p}, p), 1)$. $\square$

## C. Base Algorithm

In this section we focus on our base algorithm and analysis: we design an algorithm that uses a distributional prediction in an essentially optimal way (as the lower bounds discussed earlier show). We first give our base algorithm formally in Algorithm 3. This algorithm guarantees an upper bound

---

[6]Choose the truth $p^i$ such that the total probability mass of deterministic algorithms that perform poorly under $p^i$ is bounded below by a constant, thus holds for any randomized algorithm by Yao's Lemma.

(Theorem 8). We present the intuition of our proof (Appendix C.1), followed by a full proof of it (Appendix C.2). Without loss of generality, we assume that $\sqrt{b}$ is an integer; otherwise, we can use $\lfloor \sqrt{b} \rfloor$ instead.

---

**Algorithm 3** Base algorithm, with expected additive loss bounded by $O(\sqrt{b}\max(\text{EMD}, 1))$

---

1: **if** $A_{\hat{K}}$ is the optimal policy for $\hat{p}$ with some finite $\hat{K}$ **then**
2:     Buy at the beginning of day $\hat{K}+\sqrt{b}+1$, i.e., $ALG = A_{\hat{K}+\sqrt{b}}$
3: **else**
4:     Keep renting, i.e., $ALG = A_\infty$
5: **end if**

---

In other words, Algorithm 3 distinguishes two cases depending on the optimal policy under the predicted distribution $\hat{p}$. If the optimal policy for $\hat{p}$ is of the form $A_{\hat{K}}$ for some finite $\hat{K}$, the algorithm delays the purchase time by an additive $\sqrt{b}$ and buys at time $\hat{K}+\sqrt{b}+1$. Otherwise, when the optimal policy for $\hat{p}$ is $A_\infty$, the algorithm simply keeps renting and never buys.

Recall the statement of Theorem 8:

**Theorem 8.** *The base algorithm has expected additive loss*

$$c(ALG_p) - c(OPT_p) \leq O\left(\sqrt{b} \cdot \max\left(EMD(\hat{p}, p), 1\right)\right)$$

To prove Theorem 8, we break into cases depending on the structure of the optimal policies for $\hat{p}$ and $p$. We first prove Theorem 8 when the optimal policy for $\hat{p}$ is $A_\infty$ (Lemma 14). When the optimal policy for $\hat{p}$ is $A_K$ with finite $K$ the analysis is more complicated, and we need to break into two further cases depending on the true optimal policy for $p$. In Lemma 16 we prove the theorem for the optimal policy (for $p$) being $A_\infty$ and in Lemma 17 we consider the case when both the optimal policies for $p$ and for $\hat{p}$ involve buying at some finite time. Together, these lemmas directly imply Theorem 8.

### C.1. Intuition of Proof

While the full proof is in Appendix C.2, we now provide some intuition about it. First, it is helpful for us to break into cases depending on the optimal policies for $p$ and $\hat{p}$ (as discussed above) because once we know the optimal policy we can explicitly write down the cost of the optimal policy, which allows us to write diff$_p$ explicitly.

Now suppose that the true distribution were actually $\hat{p}$ rather than $p$. Then clearly our algorithm is optimal if this policy is $A_\infty$, and otherwise is suboptimal. But we can also consider what happens if we run $OPT_p$ (the optimal policy for $p$) when the truth is $\hat{p}$, and we can define diff$_{\hat{p}}$ as the expected

cost of our algorithm minus the expected cost of $OPT_p$ when the true distribution is $\hat{p}$. So diff$_{\hat{p}}$ is non-positive when our policy is $A_\infty$ (since in this case we are optimal), but otherwise can be quite large.

If it *were* true that diff$_{\hat{p}}$ was always small, then a natural strategy would be to start with that as a baseline and analyze how the diff$_p$ increases when compared with diff$_{\hat{p}}$. In other words, we could try to argue that diff$_p -$ diff$_{\hat{p}} \leq O(\sqrt{b}\max(EMD(\hat{p}, p), 1))$, which would then imply Theorem 8 if diff$_{\hat{p}}$ were small enough. Unfortunately this runs into two related issues. First, diff$_{\hat{p}}$ could be extremely negative, making our desired claim false. Second, diff$_{\hat{p}}$ could be extremely large and positive, making the claim true but insufficient to prove Theorem 8. Fortunately, we can get around both of these issues in the same way: we can simply normalize by diff$_{\hat{p}}$. More formally, we can prove that diff$_p -$diff$_{\hat{p}} \leq O(\sqrt{b}\max(EMD(\hat{p}, p), 1)) -$diff$_{\hat{p}}$. Simply adding diff$_{\hat{p}}$ to both sides then implies the theorem.

So our strategy is try to bound diff$_p -$ diff$_{\hat{p}}$. This is about the change in the diff function when the distribution changes from $\hat{p}$ to $p$, and so a natural approach is to utilize the optimal transport plan from $\hat{p}$ to $p$. We call each value $f(x, y)$ in this transport plan a *map*, since it tells us how much probability mass to move from $x$ to $y$. Some of the maps either benefit us (i.e., result in diff$_p$ decreasing compared to diff$_{\hat{p}}$) or result in an increase that can still be bounded in simple ways (e.g., directly bounded by $EMD(\hat{p}, p)$). We call such maps *good maps* and all others are *bad maps*, and only need concern ourselves with bad maps. Although which maps are bad depends on which precise case we are in, we generally split the bad maps into two types: maps that move mass from $t_1$ to $t_2$ with $|t_1 - t_2| \geq \sqrt{b}$ and maps that move mass from $t_1$ to $t_2$ with $|t_1 - t_2| < \sqrt{b}$. If a map is in the regime where the moving distance is large (larger than $\sqrt{b}$), the amount of mass moved by this map is then bounded by $\frac{EMD(\hat{p}, p)}{\sqrt{b}}$ by the definition of $EMD(\hat{p}, p)$. This is a small enough amount of mass that we can straightforwardly argue that the increase it causes lies within our desired bound.

If a map is in the regime where the moving distance is small (smaller than $\sqrt{b}$), then we instead directly use the fact that our policy is related to (though not identical to) the optimal policy for $\hat{p}$. In particular, an obvious observation is that when moving mass from $\hat{p}$ to $p$ using those bad maps in the optimal transport plan, the amount of mass one can move from point $t$ is at most $\hat{p}_t$. So if we can argue that $\hat{p}_t$ is small, then we get that these types of bad maps move a small amount of mass a small distance, and so the increase they cause in the diff function still lives within our desired bound. And it turns out that we can indeed show this, although the precise guarantee and strategy is different for different cases. But for example, when the optimal policy for $\hat{p}$ is $A_K$, it is intuitively true (and can

be made formal) that $\sum_{t=K}^{K+\delta} \hat{p}_t$ has to be small when $\delta$ is small (otherwise renting $K$ times then buying would not be the optimal policy; it would be better for us to wait a little longer to buy). Therefore, we leverage the mass distribution in $\hat{p}$ to connect the change in additive loss $\text{diff}_p - \text{diff}_{\hat{p}}$ with the required bound $O(\sqrt{b}\max(EMD(\hat{p}, p), 1)) - \text{diff}_{\hat{p}}$.

## C.2. Full Proof

This section gives the technical proofs of Theorem 8. we break into cases depending on the optimal policies for $\hat{p}$ and $p$ to get the claimed bound.

**Lemma 14.** *Let $\hat{p}$ be the prediction and $p$ be the unknown truth. If the optimal policy for $\hat{p}$ is $A_\infty$, then taking $ALG = A_\infty$, we have $\text{diff}_p = c(ALG_p) - c(OPT_p) \leq O(\sqrt{b}\max(EMD(\hat{p}, p), 1))$.*

*Proof.* If the optimal policy for $p$ is also $A_\infty$, then $\text{diff}_p = 0$ and the required bound holds trivially. Thus, we only need to show that the bound holds when the optimal policy for $p$ is $A_K$ for some finite $K$.

First, we have

$$\text{diff}_p = \sum_{t=1}^{N} tp_t - \left( \sum_{t \leq K} tp_t + \sum_{t > K} (K+b)p_t \right)$$

$$= \sum_{t > K} (t - (K+b))p_t.$$

Since $\text{diff}_p$ is a function of $p$, we similarly define $\text{diff}_{\hat{p}}$ as the quantity by using the same policies but the different distribution $\hat{p}$ rather than $p$. In other words, $\text{diff}_{\hat{p}} = c(ALG_{\hat{p}}(\hat{p})) - c((OPT_p)_{\hat{p}})$. Then, we consider the optimal transport plan from $\hat{p}$ to $p$ and examine the change of the additive loss $\text{diff}_p - \text{diff}_{\hat{p}}$. We want to show that $\text{diff}_p - \text{diff}_{\hat{p}} \leq O(\sqrt{b}\max(EMD(\hat{p}, p), 1)) - \text{diff}_{\hat{p}}$. That is, we want this change in additive loss to be bounded by $O(\sqrt{b}\max(EMD(\hat{p}, p), 1)) - \text{diff}_{\hat{p}}$.

Note that $A_\infty$ is the optimal policy for $\hat{p}$, we have $\text{diff}_{\hat{p}} \leq 0$ and

$$\text{diff}_{\hat{p}} = \sum_{t > K} (t - (K+b))\hat{p}_t$$

$$= \sum_{t > K+b} (t - (K+b))\hat{p}_t - \sum_{K < t < K+b} (K+b-t)\hat{p}_t.$$

We divide the domain into 3 regions: $[1, K], (K, K+b], (K+b, \infty)$. There are two kinds of maps in the optimal transport plan: from some point in $(K, K+b]$ to some point in $[1, K]$, or not in such a form. If we consider how a map from one region to any region (including itself) affects the change in additive loss, notice that for any map that is not from $(K, K+b]$ to $[1, K]$, the change in additive loss by the map is at most its own contribution to the Earth

Mover's Distance. To argue this, if a map is from $[1, K]$ to $[1, K]$, then it does not affect the change in additive loss. If a map is from $[1, K]$ to $(K, K+b)$, then it only decreases the value and hence benefits us. If a map is from $t_1 \in [1, K]$ to $t_2 \in (K+b, \infty)$ with mass $\epsilon$, then it increases the value by $(t_2 - (K+b))\epsilon < (t_2 - t_1)\epsilon$ which equals its contribution to the $EMD$. If a map is from $t_1 \in (K, K+b]$ to $t_2 \in (K, K+b]$ with mass $\epsilon$, then it changes the value by $(K+b-t_1)\epsilon - (K+b-t_2)\epsilon \leq |t_1-t_2|\epsilon$ which equals its contribution to the $EMD$. If a map is from $t_1 \in (K, K+b]$ to $t_2 \in (K+b, \infty)$ with mass $\epsilon$, then it increases the value by $(K+b-t_1)\epsilon + (t_2 - (K+b))\epsilon = (t_2 - t_1)\epsilon$ which equals its contribution to the $EMD$. If a map is from $(K+b, \infty)$ to $[1, K]$ or from $(K+b, \infty)$ to $(K, K+b]$, then again it only decreases the value and benefits us. If a map is from $t_1 \in (K+b, \infty)$ to $t_2 \in (K+b, \infty)$ with mass $\epsilon$, then it changes the value by $(t_2 - (K+b))\epsilon - (t_1 - (K+b))\epsilon = (t_2 - t_1)\epsilon$ which equals its contribution to the $EMD$. Thus, if we consider all such maps (not from $(K, K+b]$ to $[1, K]$) as a group, it will increase the additive loss by some amount bounded by $EMD(\hat{p}, p)$ which is at most a constant. We call such maps in this easy case the good maps and those we need to take further actions the bad maps.

For maps (elements in the optimal transport plan) from $(K, K+b]$ to $[1, K]$, we have $\text{diff}_p - \text{diff}_{\hat{p}} = \sum_{K < t < K+b} (K+b-t)(\hat{p}_t - p_t)$. Let us first consider the set of bad maps $\mathcal{B}_1$ from $(K+\sqrt{b}, K+b]$ to $[1, K]$. Since the moving distance for such map is at least $\sqrt{b}$, the total mass moving by such bad maps is at most $O(\frac{EMD}{\sqrt{b}})$. Thus,

$$\text{diff}_p - \text{diff}_{\hat{p}}|_{\mathcal{B}_1} \leq (K+b-(K+\sqrt{b})) \sum_{K+\sqrt{b} < t \leq K+b} \epsilon_t|_{\mathcal{B}_1}$$

$$\leq (b - \sqrt{b}) \cdot O\left( \frac{EMD}{\sqrt{b}} \right)$$

$$= O(\sqrt{b} \cdot EMD).$$

where $\epsilon_t|_{\mathcal{B}_1} = \hat{p}_t - p_t|_{\mathcal{B}_1}$ is the mass change by all such bad maps in $\mathcal{B}_1$ at $t$ for any point $K+\sqrt{b} < t \leq K+b$.

Then, let us consider the set of bad maps $\mathcal{B}_2$ from $(K, K+\sqrt{b}]$ to $[1, K]$. We will discuss the change in additive loss by such bad maps in $\mathcal{B}_2$. That is, we want to bound

$$\text{diff}_p - \text{diff}_{\hat{p}}|_{\mathcal{B}_2} = \sum_{K < t \leq K+\sqrt{b}} (K+b-t)(\hat{p}_t - p_t)|_{\mathcal{B}_2}$$

$$= \sum_{K < t \leq K+\sqrt{b}} (K+b-t)\epsilon_t|_{\mathcal{B}_2}.$$

where $\epsilon_t|_{\mathcal{B}_2} = \hat{p}_t - p_t|_{\mathcal{B}_2}$ is the mass change by all such bad maps in $\mathcal{B}_2$ at $t$ for any point $K < t \leq K+\sqrt{b}$. Note that it suffices to show that $\text{diff}_p - \text{diff}_{\hat{p}}|_{\mathcal{B}_2} \leq O(\sqrt{b}) - \text{diff}_{\hat{p}}$ to get our required result.

Recall that in $\hat{p}$, $A_\infty$ is the optimal policy. Thus, we have

$$c\big((A_\infty)_{\hat{p}}\big) \leq c\big((A_{K+\sqrt{b}})_{\hat{p}}\big)$$

$$\iff \sum_t t\hat{p}_t \leq \sum_{t \leq K+\sqrt{b}} t\hat{p}_t + \sum_{t > K+\sqrt{b}} (K+\sqrt{b}+b)\hat{p}_t$$

$$\iff \sum_{t > K+\sqrt{b}} t\hat{p}_t \leq \sum_{t > K+\sqrt{b}} (K+\sqrt{b}+b)\hat{p}_t$$

$$\iff \sum_{t > K+b} \big(t - (K+\sqrt{b}+b)\big)\hat{p}_t$$

$$\leq \sum_{\substack{K+\sqrt{b} < t \\ t \leq K+b}} (K+\sqrt{b}+b-t)\hat{p}_t$$

$$\iff \sum_{t > K+b} (t-(K+b))\hat{p}_t$$

$$\leq \sum_{t > K+b} \sqrt{b}\,\hat{p}_t + \sum_{\substack{K+\sqrt{b} < t \\ t \leq K+b}} (K+\sqrt{b}+b-t)\hat{p}_t.$$

$$(\dagger)$$

By the definition of $\epsilon_t|_{\mathcal{B}_2}$, we know that $\epsilon_t|_{\mathcal{B}_2} \leq \hat{p}_t$ for $K < t \leq K+\sqrt{b}$. Therefore, to show that $\mathrm{diff}_p - \mathrm{diff}_{\hat{p}}|_{\mathcal{B}_2} \leq O(\sqrt{b}) - \mathrm{diff}_{\hat{p}} = O(\sqrt{b}) - (\sum_{t>K}(t-(K+b))\hat{p}_t)$, it suffices to show that

$$\sum_{K < t \leq K+\sqrt{b}} (K+b-t)\hat{p}_t \leq O(\sqrt{b}) - \sum_{t>K} (t-(K+b))\hat{p}_t$$

which is equivalently

$$\sum_{t > K+b} (t-(K+b))\hat{p}_t \leq O(\sqrt{b}) + \sum_{K+\sqrt{b} < t \leq K+b} (K+b-t)\hat{p}_t.$$

The last line holds because of ($\dagger$).

By combining results from good maps and bad maps, we finally get the required result. $\qquad\square$

Next, we show that when the optimal policy for $\hat{p}$ is $A_K$ for some finite $K$, taking $ALG = A_{K+\sqrt{b}}$ implies the required bound. Before proving this, let us make some observations.

**Observation 1.** $c\big((A_{K+\sqrt{b}})_{\hat{p}}\big) \leq c\big((A_K)_{\hat{p}}\big) + \sqrt{b}$.

*Proof.* We write down the definition of $c\big((A_{K+\sqrt{b}})_{\hat{p}}\big)$ and $c\big((A_K)_{\hat{p}}\big)$. It is not hard to see that by taking the difference we have $c\big((A_{K+\sqrt{b}})_{\hat{p}}\big) - c\big((A_K)_{\hat{p}}\big) \leq \sqrt{b}\sum_{t>K+\sqrt{b}}\hat{p}_t \leq \sqrt{b}$. $\qquad\square$

**Claim 15.** *Let $A_K$ be the optimal policy for $\hat{p}$. Then, for every $1 \leq T \leq b$, we have*

$$\sum_{K < t \leq K+T} \hat{p}_t \leq \frac{T}{b} \sum_{t>K} \hat{p}_t.$$

*Proof.* For $1 \leq T \leq b$, we will compare the cost of $A_K$ and $A_{K+T}$.

$$c\big((A_K)_{\hat{p}}\big) \leq c\big((A_{K+T})_{\hat{p}}\big)$$

$$\iff \sum_{t \leq K} t\hat{p}_t + \sum_{t>K}(K+b)\hat{p}_t$$

$$\leq \sum_{t \leq K+T} t\hat{p}_t + \sum_{t > K+T}(K+T+b)\hat{p}_t$$

$$\iff \sum_{K < t \leq K+T}(K+b)\hat{p}_t \leq \sum_{K < t \leq K+T} t\hat{p}_t + \sum_{t > K+T} T\hat{p}_t$$

$$\iff \sum_{K < t \leq K+T}(K+b-t)\hat{p}_t \leq \sum_{t > K+T} T\hat{p}_t.$$

Thus, we have

$$b \sum_{K < t \leq K+T} \hat{p}_t \leq T \sum_{K < t \leq K+T} \hat{p}_t + \sum_{K < t \leq K+T}(K+b-t)\hat{p}_t$$

$$\leq T \sum_{K < t \leq K+T} \hat{p}_t + T \sum_{t > K+T} \hat{p}_t = T \sum_{t>K} \hat{p}_t.$$

That is, $\sum_{K < t \leq K+T} \hat{p}_t \leq \frac{T}{b}\sum_{t>K}\hat{p}_t$. $\qquad\square$

Now, we can start with the proof. In the following lemmas, we will analyze the cases separately based on the optimal policy for $p$ to be either $A_\infty$ or some $A_L$, given that the optimal policy for $\hat{p}$ is $A_K$ for some finite $K$ as well as $ALG = A_{K+\sqrt{b}}$ always.

**Lemma 16.** *Let $\hat{p}$ be the prediction and $p$ be the unknown truth. Assume that the optimal policy for $\hat{p}$ is $A_K$ for some finite $K$. Let $ALG = A_{K+\sqrt{b}}$. If the optimal policy for $p$ is $A_\infty$, we have $\mathrm{diff}_p = c(ALG_p) - c(OPT_p) \leq O(\sqrt{b}\max(EMD(\hat{p},p),1))$.*

*Proof.* We want to show that $\mathrm{diff}_p = c\big((A_{K+\sqrt{b}})_p\big) - c\big((A_\infty)_p\big) \leq O(\sqrt{b}\max(EMD(\hat{p},p),1))$. Consider $\mathrm{diff}_{\hat{p}} = c\big((A_{K+\sqrt{b}})_{\hat{p}}\big) - c\big((A_\infty)_{\hat{p}}\big)$. Since $A_K$ is the optimal policy for $\hat{p}$, we have $c\big((A_K)_{\hat{p}}\big) - c\big((A_\infty)_{\hat{p}}\big) \leq 0$. By Observation 1, we have

$$\mathrm{diff}_{\hat{p}} = c\big((A_{K+\sqrt{b}})_{\hat{p}}\big) - c\big((A_\infty)_{\hat{p}}\big)$$

$$\leq c\big((A_K)_{\hat{p}}\big) + \sqrt{b} - c\big((A_\infty)_{\hat{p}}\big)$$

$$\leq \sqrt{b}.$$

It turns out that if we can show $\mathrm{diff}_p - \mathrm{diff}_{\hat{p}} \leq O(\sqrt{b}\max(EMD(\hat{p},p),1))$, then $\mathrm{diff}_p - \mathrm{diff}_{\hat{p}} \leq O(\sqrt{b}\max(EMD(\hat{p},p),1)) - \mathrm{diff}_{\hat{p}}$ and we are done.

To show that $\mathrm{diff}_p - \mathrm{diff}_{\hat{p}} \leq O(\sqrt{b}\max(EMD(\hat{p},p),1))$, again we use the optimal transport plan and analyze the bad

maps. After simple calculations, we have

$$\text{diff}_{\hat{p}} = \sum_{K+\sqrt{b}<t\le K+\sqrt{b}+b} (K + \sqrt{b} + b - t)\,\hat{p}_t$$
$$- \sum_{t>K+\sqrt{b}+b} \left(t - (K + \sqrt{b} + b)\right)\hat{p}_t.$$

Then, by considering maps between the regions $[1, K + \sqrt{b}]$, $(K + \sqrt{b}, K + \sqrt{b} + b]$ and $(K + \sqrt{b} + b, \infty)$, the bad maps are those from $[1, K + \sqrt{b}]$ to $(K + \sqrt{b}, K + \sqrt{b} + b]$ (other maps are again digested by their contributions to the $EMD$, as we did in Lemma 14).

For maps $\mathcal{B}$ from $[1, K + \sqrt{b}]$ to $(K + \sqrt{b}, K + \sqrt{b} + b]$ (elements in the optimal transport plan), since $\mathcal{B}$ has nothing to do with $t > K + \sqrt{b} + b$, we have

$$\text{diff}_p - \text{diff}_{\hat{p}}|_{\mathcal{B}} = \sum_{K+\sqrt{b}<t\le K+\sqrt{b}+b} (K+\sqrt{b}+b-t)(p_t-\hat{p}_t).$$

Let us first consider the set of bad maps $\mathcal{B}_1$ from $[1, K]$ to $(K + \sqrt{b}, K + \sqrt{b} + b]$. Since the moving distance for such map is at least $\sqrt{b}$, the total mass moving by such bad maps is at most $O(\frac{EMD}{\sqrt{b}})$. Thus,

$$\text{diff}_p - \text{diff}_{\hat{p}}|_{\mathcal{B}_1} = \sum_{\substack{K+\sqrt{b}<t \\ t\le K+\sqrt{b}+b}} (K + \sqrt{b} + b - t)\,\epsilon_t|_{\mathcal{B}_1}$$
$$\le b \cdot \left| \sum_{K+\sqrt{b}<t\le K+\sqrt{b}+b} \epsilon_t \right|_{\mathcal{B}_1}$$
$$\le O(\sqrt{b} \cdot EMD)$$

where $\epsilon_t|_{\mathcal{B}_1} = p_t - \hat{p}_t|_{\mathcal{B}_1}$ is the mass change by all such bad maps in $\mathcal{B}_1$ at $t$ for any point $K+\sqrt{b} < t \le K+\sqrt{b}+b$.

Then, let us consider the set of bad maps $\mathcal{B}_2$ from $(K, K + \sqrt{b}]$ to $(K + \sqrt{b}, K + \sqrt{b} + b]$. We similarly define $\epsilon_t|_{\mathcal{B}_2} = p_t - \hat{p}_t|_{\mathcal{B}_2}$ as the mass change by all such bad maps in $\mathcal{B}_2$ at $t$ for any point $K + \sqrt{b} < t \le K + \sqrt{b} + b$. Since we are considering bad maps of the form $\mathcal{B}_2$, we know that $\sum_{K+\sqrt{b}<t\le K+\sqrt{b}+b} \epsilon_t\big|_{\mathcal{B}_2}$ is at most $\sum_{K<t\le K+\sqrt{b}} \hat{p}_t$. Therefore,

$$\text{diff}_p - \text{diff}_{\hat{p}}\big|_{\mathcal{B}_2} = \sum_{\substack{K+\sqrt{b}<t \\ t\le K+\sqrt{b}+b}} (K + \sqrt{b} + b - t)\,\epsilon_t\big|_{\mathcal{B}_2}$$
$$\le b \sum_{K+\sqrt{b}<t\le K+\sqrt{b}+b} \epsilon_t\big|_{\mathcal{B}_2}$$
$$\le b \sum_{K<t\le K+\sqrt{b}} \hat{p}_t.$$

Again, this is bounded by $O(\sqrt{b})$ by taking $T = \sqrt{b}$ in Claim 15.

Therefore, we finish the proof as required. $\qquad\square$

Now, the only left case is when $A_{\hat{K}}$ is the optimal policy for $\hat{p}$ and $A_K$ is the optimal policy for $p$ for some finite $\hat{K}$ and $K$. In this case, we refer to the optimal policy for $\hat{p}$ as $A_{\hat{K}}$ to avoid any ambiguity. We need to show that by taking $ALG = A_{\hat{K}+\sqrt{b}}$, we have $\text{diff}_p \le O(\sqrt{b}\max(EMD(\hat{p},p),1))$. This needs a long proof, as one will see that we have to use a combination of techniques in the previous lemmas.

**Lemma 17.** *Let $\hat{p}$ be the prediction and $p$ be the truth. Assume that the optimal policy for $\hat{p}$ is $A_{\hat{K}}$ for some finite $\hat{K}$. Let $ALG = A_{\hat{K}+\sqrt{b}}$. If the optimal policy for $p$ is $A_K$ for some finite $K$, we have $\text{diff}_p = c(ALG_p) - c(OPT_p) \le O(\sqrt{b}\max(EMD(\hat{p},p),1))$.*

*Proof.* We want to show that $\text{diff}_p = c((A_{\hat{K}+\sqrt{b}})_p) - c((A_K)_p) \le O(\sqrt{b})$. Consider $\text{diff}_{\hat{p}} = c((A_{\hat{K}+\sqrt{b}})_{\hat{p}}) - c((A_K)_{\hat{p}})$. Since $A_{\hat{K}}$ is the optimal policy for $\hat{p}$, we have $c((A_{\hat{K}})_{\hat{p}}) - c((A_K)_{\hat{p}}) \le 0$. By Observation 1, we have

$$\text{diff}_{\hat{p}} = c\left((A_{\hat{K}+\sqrt{b}})_{\hat{p}}\right) - c\left((A_K)_{\hat{p}}\right)$$
$$\le c\left((A_{\hat{K}})_{\hat{p}}\right) + \sqrt{b} - c\left((A_K)_{\hat{p}}\right)$$
$$\le \sqrt{b}.$$

Now, we want to find the bad maps in an optimal transport plan. We write down

$$\text{diff}_{\hat{p}} = \left[ \sum_{t\le\hat{K}+\sqrt{b}} t\,\hat{p}_t + \sum_{t>\hat{K}+\sqrt{b}} (\hat{K}+\sqrt{b}+b)\,\hat{p}_t \right]$$
$$- \left[ \sum_{t\le K} t\,\hat{p}_t + \sum_{t>K} (K+b)\,\hat{p}_t \right].$$

We separate into two cases by comparing $\hat{K} + \sqrt{b}$ and $K$.

**Case 1.** When $\hat{K} + \sqrt{b} < K$.

In this case, by some simple calculations, we have

$$\text{diff}_{\hat{p}} = \sum_{\substack{\hat{K}+\sqrt{b}<t \\ t\le K}} (\hat{K}+\sqrt{b}+b-t)\hat{p}_t$$
$$- \sum_{t>K} (K - (\hat{K}+\sqrt{b}))\hat{p}_t$$
$$= \sum_{t\le\hat{K}+\sqrt{b}} \left(K - (\hat{K}+\sqrt{b})\right)\hat{p}_t$$
$$+ \sum_{\substack{\hat{K}+\sqrt{b}<t \\ t\le K}} (K+b-t)\,\hat{p}_t - \left(K - (\hat{K}+\sqrt{b})\right).$$

By examining different kind of maps, one can check that there are two kinds of bad maps: either from $[1, \hat{K} + \sqrt{b}]$ to $(\hat{K} + \sqrt{b}, K]$ (bad maps of the first kind $\mathcal{B}^1$), or from

$(K, K + b]$ to $(\hat{K} + \sqrt{b}, K]$ (bad maps of the second kind $\mathcal{B}^2$). We will analyze those bad maps of the first kind using the first expression of $\mathrm{diff}_{\hat{p}}$ and analyze those bad maps of the second kind using the second expression in the above formula.

For the set of bad maps of the first kind which are from $[1, \hat{K}]$ to $(\hat{K} + \sqrt{b}, K]$, we denote it as $\mathcal{B}_1^1$. We know that maps in $\mathcal{B}_1^1$ has nothing to do with $t > K$. Moreover, since the moving distance for such map is at least $\sqrt{b}$, the total mass moving by such bad maps is at most $O(\frac{EMD}{\sqrt{b}})$. Therefore,

$$
\begin{aligned}
\mathrm{diff}_p - \mathrm{diff}_{\hat{p}}|_{\mathcal{B}_1^1} &= \sum_{\substack{\hat{K}+\sqrt{b}<t\leq K}} (\hat{K} + \sqrt{b} + b - t)(p_t - \hat{p}_t)|_{\mathcal{B}_1^1} \\
&\leq b \sum_{\hat{K}+\sqrt{b}<t\leq K} \epsilon_t\big|_{\mathcal{B}_1^1} \\
&\leq O\left(\sqrt{b} \cdot EMD\right),
\end{aligned}
$$

where $\epsilon_t|_{\mathcal{B}_1^1} = p_t - \hat{p}_t|_{\mathcal{B}_1^1}$ is the mass change by all such bad maps in $\mathcal{B}_1^1$ at $t$ for any point $\hat{K} + \sqrt{b} < t \leq K$.

Then, let us consider the set of bad maps $\mathcal{B}_2^1$ from $(\hat{K}, \hat{K} + \sqrt{b}]$ to $(\hat{K} + \sqrt{b}, K]$. We similarly define $\epsilon_t|_{\mathcal{B}_2^1} = p_t - \hat{p}_t|_{\mathcal{B}_2^1}$ as the mass change by all such bad maps in $\mathcal{B}_2^1$ at $t$ for any point $\hat{K} + \sqrt{b} < t \leq K$.

Since we are considering bad maps of the form $\mathcal{B}_2^1$, we know that $\sum_{\hat{K}+\sqrt{b}<t\leq K} \epsilon_t\big|_{\mathcal{B}_2^1}$ is at most $\sum_{\hat{K}<t\leq \hat{K}+\sqrt{b}} \hat{p}_t$. Therefore,

$$
\begin{aligned}
\mathrm{diff}_p - \mathrm{diff}_{\hat{p}}\big|_{\mathcal{B}_2^1} &= \sum_{\hat{K}+\sqrt{b}<t\leq K} (\hat{K} + \sqrt{b} + b - t)\,\epsilon_t\big|_{\mathcal{B}_2^1} \\
&\leq b \sum_{\hat{K}+\sqrt{b}<t\leq K} \epsilon_t\big|_{\mathcal{B}_2^1} \\
&\leq b \sum_{\hat{K}<t\leq \hat{K}+\sqrt{b}} \hat{p}_t.
\end{aligned}
$$

Again, this is bounded by $O(\sqrt{b})$ by taking $T = \sqrt{b}$ in Claim 15.

For the set of bad maps of the second kind which are from $(K + \sqrt{b}, K + b]$ to $(\hat{K} + \sqrt{b}, K]$, we denote it as $\mathcal{B}_1^2$. We know that maps in $\mathcal{B}_1^2$ has nothing to do with $t < \hat{K} + \sqrt{b}$. Moreover, since the moving distance for such map is at least $\sqrt{b}$, the total mass moving by such bad maps is at most

$O(\frac{EMD}{\sqrt{b}})$. Therefore,

$$
\begin{aligned}
\mathrm{diff}_p - \mathrm{diff}_{\hat{p}}\big|_{\mathcal{B}_1^2} &= \sum_{\substack{\hat{K}+\sqrt{b}<t \\ t\leq K}} (K + b - t)\,(p_t - \hat{p}_t)\big|_{\mathcal{B}_1^2} \\
&\leq \sum_{\substack{\hat{K}+\sqrt{b}<t \\ t\leq K}} (K - t)\,\epsilon_t\big|_{\mathcal{B}_1^2} + \sum_{\substack{\hat{K}+\sqrt{b}<t \\ t\leq K}} b\,\epsilon_t\big|_{\mathcal{B}_1^2} \\
&\leq O\left(\sqrt{b} \cdot EMD\right),
\end{aligned}
$$

where $\epsilon_t|_{\mathcal{B}_1^2} = p_t - \hat{p}_t|_{\mathcal{B}_1^2}$ is the mass change by all such bad maps in $\mathcal{B}_1^2$ at $t$ for any point $\hat{K} + \sqrt{b} < t \leq K$. The last inequality holds since $\sum_{\hat{K}+\sqrt{b}<t\leq K}(K - t)\,\epsilon_t|_{\mathcal{B}_1^2}$ is bounded by $EMD(\hat{p}, p)$ and $\sum_{\hat{K}+\sqrt{b}<t\leq K} b\,\epsilon_t|_{\mathcal{B}_1^2} \leq O(b \cdot \frac{EMD}{\sqrt{b}}) = O(\sqrt{b} \cdot EMD)$.

Then, let us consider the set of bad maps $\mathcal{B}_2^2$ from $(K, K + \sqrt{b}]$ to $(\hat{K} + \sqrt{b}, K]$. This is a bit complicated. First, recall that in $\hat{p}$, $A_{\hat{K}}$ is the optimal policy, so $c((A_{\hat{K}})_{\hat{p}}) \leq c((A_{K+\sqrt{b}})_{\hat{p}})$ (note that in case 1, $\hat{K} < K + \sqrt{b}$). We have

$$
\begin{aligned}
& c\big((A_{\hat{K}})_{\hat{p}}\big) \leq c\big((A_{K+\sqrt{b}})_{\hat{p}}\big) \\
\Longleftrightarrow\ & \sum_{t\leq K+\sqrt{b}} (K + \sqrt{b} + b - t)\,\hat{p}_t \\
& \leq (K + \sqrt{b} - \hat{K}) + \sum_{t\leq \hat{K}} (\hat{K} + b - t)\,\hat{p}_t \\
\Longleftrightarrow\ & \sum_{K<t\leq K+\sqrt{b}} (K + \sqrt{b} + b - t)\,\hat{p}_t \\
& \leq (K + \sqrt{b} - \hat{K}) + \sum_{t\leq \hat{K}} (\hat{K} + b - t)\,\hat{p}_t \\
& \quad - \sum_{t\leq K} (K + \sqrt{b} + b - t)\,\hat{p}_t. \qquad (*)
\end{aligned}
$$

We similarly define $\epsilon_t|_{\mathcal{B}_2^2} = p_t - \hat{p}_t|_{\mathcal{B}_2^2}$ as the mass change by all such bad maps in $\mathcal{B}_2^2$ at $t$ for any point $\hat{K} + \sqrt{b} < t \leq K$.

Since we are considering bad maps of the form $\mathcal{B}_2^2$, we know that $\sum_{\hat{K}+\sqrt{b}<t\leq K} \epsilon_t\big|_{\mathcal{B}_2^2}$ is at most $\sum_{K<t\leq K+\sqrt{b}} \hat{p}_t$. Therefore,

$$
\begin{aligned}
\big(\mathrm{diff}_p - \mathrm{diff}_{\hat{p}}\big)\big|_{\mathcal{B}_2^2} &= \sum_{\hat{K}+\sqrt{b}<t\leq K} (K + b - t)\,\epsilon_t\big|_{\mathcal{B}_2^2} \\
&\leq \sum_{\hat{K}+\sqrt{b}<t\leq K} (K - t)\,\epsilon_t\big|_{\mathcal{B}_2^2} \\
&\quad + \sum_{\hat{K}+\sqrt{b}<t\leq K} b\,\epsilon_t\big|_{\mathcal{B}_2^2}.
\end{aligned}
$$

Note that term $\sum_{\hat{K}+\sqrt{b}<t\leq K}(K - t)\,\epsilon_t|_{\mathcal{B}_2^2}$ is bounded by $EMD(\hat{p}, p)$ and $\sum_{\hat{K}+\sqrt{b}<t\leq K} b\,\epsilon_t|_{\mathcal{B}_2^2}$ is bounded by

$b \sum_{K < t \le K + \sqrt{b}} \hat{p}_t$. Thus, to show that $\text{diff}_p - \text{diff}_{\hat{p}} \le O(\sqrt{b} \cdot \max(EMD, 1)) - \text{diff}_{\hat{p}}$, it suffices to show that $b \sum_{K < t \le K + \sqrt{b}} \hat{p}_t \le O(\sqrt{b}) - \text{diff}_{\hat{p}}$.

Note that $b \sum_{K < t \le K + \sqrt{b}} \hat{p}_t$ is less than the left hand side of $(*)$, hence it suffices to show that

$$(K + \sqrt{b} - \hat{K}) + \sum_{t \le \hat{K}} (\hat{K} + b - t)\hat{p}_t$$
$$- \sum_{t \le K} (K + \sqrt{b} + b - t)\hat{p}_t \le O(\sqrt{b}) - \text{diff}_{\hat{p}}.$$

By writing $\text{diff}_{\hat{p}} = \sum_{t \le \hat{K} + \sqrt{b}} (K - (\hat{K} + \sqrt{b}))\hat{p}_t + \sum_{\hat{K} + \sqrt{b} < t \le K} (K + b - t)\hat{p}_t - (K - (\hat{K} + \sqrt{b}))$, it suffices to show that (rearranging to make both sides containing positive terms only)

$$\sum_{t \le \hat{K}} (\hat{K} + b - t)\hat{p}_t + \sum_{t \le \hat{K} + \sqrt{b}} \left(K - (\hat{K} + \sqrt{b})\right)\hat{p}_t$$
$$+ \sum_{\hat{K} + \sqrt{b} < t \le K} (K + b - t)\hat{p}_t \qquad (\S\S)$$
$$\le O(\sqrt{b}) + \sum_{t \le K} (K + \sqrt{b} + b - t)\hat{p}_t.$$

Now, compare both sides of $(\S\S)$ for all $t$.

For $t \le \hat{K}$, $LHS$ of $(\S\S)$ is $(\hat{K} + b - t + K - (\hat{K} + \sqrt{b}))\hat{p}_t = (b + K - t - \sqrt{b})\hat{p}_t$. $RHS$ of $(\S\S)$ is $(K + \sqrt{b} + b - t)\hat{p}_t$. So, $LHS \le RHS$.

For $\hat{K} < t \le \hat{K} + \sqrt{b}$, $LHS$ of $(\S\S)$ is $(K - (\hat{K} + \sqrt{b}))\hat{p}_t$. $RHS$ of $(\S\S)$ is $(K + \sqrt{b} + b - t)\hat{p}_t$. So, $LHS \le RHS$ since $t \le \hat{K} + \sqrt{b}$.

For $\hat{K} + \sqrt{b} < t \le K$, $LHS$ of $(\S\S)$ is $(K + b - t)\hat{p}_t$. $RHS$ of $(\S\S)$ is $(K + \sqrt{b} + b - t)\hat{p}_t$. So, $LHS \le RHS$.

Therefore, $(\S\S)$ holds. We proved that $\text{diff}_p \le O(\sqrt{b} \cdot \max(EMD, 1))$ and finish the proof of Case 1.

**Case 2.** When $K < \hat{K} + \sqrt{b}$.

We will briefly give the short proof here as the full procedure is similar to Case 1.

In this case, by some simple calculations, we have

$$\text{diff}_{\hat{p}} = \sum_{t > \hat{K} + \sqrt{b}} (\hat{K} + \sqrt{b} - K)\hat{p}_t - \sum_{\substack{K < t \\ t \le \hat{K} + \sqrt{b}}} (K + b - t)\hat{p}_t$$
$$= (\hat{K} + \sqrt{b} - K) - \sum_{t \le K} (\hat{K} + \sqrt{b} - K)\hat{p}_t$$
$$- \sum_{K < t \le \hat{K} + \sqrt{b}} (\hat{K} + \sqrt{b} + b - t)\hat{p}_t.$$

By examining different kind of maps, one can check that there are two kinds of bad maps: either from $(\hat{K}, \hat{K} + \sqrt{b}]$ to

$[1, K]$ (bad maps of the third kind $\mathcal{B}^3$), or from $(\hat{K}, \hat{K} + \sqrt{b}]$ to $(\hat{K} + \sqrt{b}, \hat{K} + \sqrt{b} + b]$ (bad maps of the fourth kind $\mathcal{B}^4$). We will analyze those bad maps of the third kind using the first expression of $\text{diff}_{\hat{p}}$ and analyze those bad maps of the fourth kind using the second expression in the above formula.

As always, we can define $\mathcal{B}_1^3$ and $\mathcal{B}_1^4$ respectively for those cases since the moving distance is at least $\sqrt{b}$ by these bad maps. We know that the total mass moving by such bad maps is at most $O(\frac{EMD}{\sqrt{b}})$, thus we have the $O(\sqrt{b} \cdot EMD)$ bound.

For those maps with a moving distance less than $\sqrt{b}$ by bad maps, we usually denote them as $\mathcal{B}_2^3$ and $\mathcal{B}_2^4$ in previous arguments. When writing down the expression $\text{diff}_p - \text{diff}_{\hat{p}}$ caused by $\mathcal{B}_2^3$ and $\mathcal{B}_2^4$, bad maps of the form $\mathcal{B}_2^4$ lie in the easier case. One can again use Claim 15 to get the required $O(\sqrt{b} \cdot EMD)$ bound, by taking $T = \sqrt{b}$. As for bad maps in $\mathcal{B}_2^3$, we consider $c((A_{\hat{K}})_{\hat{p}}) \le c((A_{K+\sqrt{b}})_{\hat{p}})$. Then $(*)$ holds in exactly the same way. Then, using $(*)$ and $b \sum_{K < t \le K + \sqrt{b}} \hat{p}_t$ as a bridge, we can write down the similar form as $(\S\S)$. Finally, we can conclude that $\text{diff}_p - \text{diff}_{\hat{p}} \le O(\sqrt{b} \cdot \max(EMD, 1)) - \text{diff}_{\hat{p}}$. This finishes the proof of Case 2 (hence the lemma). $\square$

*Proof of Theorem 8.* Theorem 8 follows directly from the combination of Lemma 14 to Lemma 17. $\square$

## D. Distributional Prediction with a Point Truth

In this section we prove the following theorem:

**Theorem 18.** *If $p$ is a point distribution, then there is a deterministic algorithm ALG which takes a distributional prediction $\hat{p}$ and has $c(ALG_p) - c(OPT_p) \le O(EMD(\hat{p}, p))$.*

In other words, we will give a new algorithm that achieves the same bound as in Purohit et al. (2018) deterministically without sampling, i.e., we derandomize the point prediction based algorithm.

First, we give Algorithm 4, in the setting that we have a distributional prediction $\hat{p}$ and a point truth $T$.

Algorithm 4 distinguishes cases according to the structure of the optimal policy under the predicted distribution $\hat{p}$. If the optimal policy for $\hat{p}$ is to buy at the beginning or to keep renting, the algorithm follows this policy. Otherwise, the optimal policy is of the form $A_K$ for some finite $K$. When $K > b$, the algorithm chooses between $A_0$ and $A_\infty$ depending on the prefix mass of $\hat{p}$ up to time $b$. When $K \le b$, the algorithm compares the prefix mass up to $K$ against a fixed threshold and either keeps renting or uses $A_K$.

Note that in the point-truth setting, the EMD can be written

**Algorithm 4** Deterministic algorithm with additive loss $O(\text{EMD})$ in the $\hat{p}$–$T$ (point truth) setting

1: **if** the optimal policy for $\hat{p}$ is $A_0$ **then**
2:     Buy at the beginning, i.e., $ALG = A_0$
3: **else if** the optimal policy for $\hat{p}$ is $A_\infty$ **then**
4:     Keep renting, i.e., $ALG = A_\infty$
5: **else**
6:     **if** $K \geq b$ **then**
7:         **if** $\sum_{t \leq b} \hat{p}_t \geq \frac{1}{3}$ **then**
8:             Keep renting, i.e., $ALG = A_\infty$
9:         **else**
10:            Buy at the beginning, i.e., $ALG = A_0$
11:         **end if**
12:     **else**
13:         **if** $\sum_{t \leq K} \hat{p}_t \geq 0.025$ **then**
14:            Keep renting, i.e., $ALG = A_\infty$
15:         **else**
16:            Buy at the beginning of day $K+1$, i.e., $ALG = A_K$
17:         **end if**
18:     **end if**
19: **end if**

as

$$EMD(\hat{p}, T) = \sum_t |t - T|\hat{p}_t = \sum_{t < T}(T - t)\hat{p}_t + \sum_{t > T}(t - T)\hat{p}_t,$$

where $\hat{p}$ is our prediction and $T$ is the truth. Since in this appendix the truth is always a point $T$, when there is no ambiguity, we will simply denote $(A_K)_T$ as $A_K$ (also for $A_0$ and $A_\infty$).

For sake of completeness, we give an example that blindly following the optimal policy for prediction $\hat{p}$ as in Purohit et al. (2018) does not perform well when the truth is a point.

**Example 1.** Let $\hat{p}_t = \begin{cases} 1 - 2^{-n}, & \text{if } t = b - 1, \\ 2^{-n}, & \text{if } t = 2b \end{cases}$ where $n$ is sufficiently large and $T = b + 1$. Since $c(A_{b-1}) = (b-1)(1 - 2^{-n}) + (2b - 1) \cdot 2^{-n} = b - 1 + b \cdot 2^{-n} < b = c(A_0)$ when $n > log(b)$, it turns out that the optimal policy for $\hat{p}$ is $A_{b-1}$. Blindly using $A_{b-1}$ when the truth $T = b + 1$ costs $2b - 1$, while the optimal cost is $b$ as $T \geq b$. Thus, the difference is $b - 1$. On the other hand, one can compute that $EMD = 2(1 - 2^{-n}) + (b - 1) \cdot 2^{-n} < 2 + (b - 1) \cdot \frac{1}{b} < 3$ is a constant. Thus, diff $\geq \Omega(b \cdot EMD)$.

Algorithm 4 follows the prediction when the optimal policy for $\hat{p}$ is either $A_0$ or $A_\infty$. We have the following lemma.

**Lemma 19.** *Assume that $\hat{p}$ is the distributional prediction and $T$ is the unknown truth. If the optimal policy for both $\hat{p}$ is restricted to be either $A_0$ or $A_\infty$, then by taking $ALG = OPT_{\hat{p}}$ we have $\text{diff}_T = c(ALG_T) - c(OPT_T) \leq EMD(\hat{p}, T)$.*

*Proof.* For simplicity, in the proof we generalize the point truth $T$ to a distributional truth $p$ whose optimal policy is also either $A_0$ or $A_\infty$. Thus, our statement is a special case, since the optimal policy for the point truth is either $A_0$ or $A_\infty$.

It is trivial when the optimal policies for both $\hat{p}$ and $p$ are $A_0$ or both are $A_\infty$, as we have $\text{diff}_p = 0$. Thus, let us consider the case where the optimal policy for $\hat{p}$ is $A_0$ while that for $p$ is $A_\infty$, and vice versa.

When the optimal policy for $\hat{p}$ is $A_0$ while that for $p$ is $A_\infty$, we have $\text{diff}_p = b - \sum_t tp_t$. Note that under $\hat{p}$, $b \leq \sum_t t\hat{p}_t$. Thus, $\text{diff}_p \leq \sum_t t\hat{p}_t - \sum_t tp_t \leq EMD(\hat{p}, p)$, where the last inequality holds by the centroid lower bound of the $EMD$. For readers who do not familiar with the centroid lower bound, a proof for centroid lower bound of the $EMD$ is deferred to Appendix F.

When the optimal policy for $\hat{p}$ is $A_\infty$ while that for $p$ is $A_0$, we have $\text{diff}_p = \sum_t tp_t - b$. Note that under $\hat{p}$, $\sum_t t\hat{p}_t \leq b$. Thus, $\text{diff}_p \leq \sum_t tp_t - \sum_t t\hat{p}_t \leq EMD(\hat{p}, p)$, where the last inequality again holds by the centroid lower bound. $\square$

Lemma 19 implies that the additive loss can be bounded by $EMD(\hat{p}, T)$ when the optimal policy for $\hat{p}$ is either $A_0$ or $A_\infty$ and we blindly follow it. Thus, the only thing left is to show that Theorem 18 holds when $A_K$ is the optimal policy for $\hat{p}$ for some $K \in \mathbb{N}_+$. First, we consider the case where $K \geq b$. We have the following lemma.

**Lemma 20.** *Assume that $A_K$ is the optimal policy for $\hat{p}$ where $K \geq b$. Consider $\sum_{t \leq b} \hat{p}_t$, i.e. the total mass in the $b$-prefix of $\hat{p}_t$. If $\sum_{t \leq b} \hat{p}_t \geq \frac{1}{3}$, we set $ALG = A_\infty$. Otherwise, we set $ALG = A_0$. Then, $\text{diff}_T = c(ALG_T) - c(OPT_T) \leq O(EMD(\hat{p}, T))$.*

*Proof.* Since $A_K$ is the optimal policy for $\hat{p}$, we have $c(A_K) \leq c(A_0)$. Thus,

$$c(A_K) \leq b$$
$$\iff \sum_{t \leq K} t\hat{p}_t + \sum_{t > K}(K + b)\hat{p}_t \leq b$$
$$\iff \sum_{t \leq K} t\hat{p}_t + (K + b) - (K + b)\sum_{t \leq K} \hat{p}_t \leq b$$
$$\iff K \leq \sum_{t \leq K}(K + b - t)\hat{p}_t$$
$$\iff K \leq \sum_{t \leq b}(K + b - t)\hat{p}_t + \sum_{b < t \leq K}(K + b - t)\hat{p}_t.$$

Thus, $K \leq (K + b)\sum_{t \leq b} \hat{p}_t + K\sum_{b < t \leq K} \hat{p}_t \leq 2K\sum_{t \leq b} \hat{p}_t + K\sum_{b < t \leq K} \hat{p}_t$. The first inequality holds since $K + b - t < K$ when $b < t$, and the second holds since $K \geq b$. Dividing $K$ on both sides, we get $1 \leq 2\sum_{t \leq b} \hat{p}_t + \sum_{b < t \leq K} \hat{p}_t$.

The above inequality guarantees that $\sum_{t \leq b} \hat{p}_t \geq \frac{1}{3}$ or $\sum_{b < t \leq K} \hat{p}_t \geq \frac{1}{3}$, or both. Now, if $\sum_{t \leq b} \hat{p}_t \geq \frac{1}{3}$, we have $ALG = A_\infty$. Otherwise, it must be $\sum_{b < t \leq K} \hat{p}_t \geq \frac{1}{3}$, we have $ALG = A_0$.

For $\sum_{t \leq b} \hat{p}_t \geq \frac{1}{3}$, if $T \leq b$, then our $ALG$ is optimal and $\text{diff}_T$ is zero. If $T > b$, $c(ALG_T) = T$ and $c(OPT_T) = b$, then $\text{diff}_T = T - b$. Thus, we have $\text{diff}_T = T - b \leq O(EMD(\hat{p}, T))$ because $EMD(\hat{p}, T) \geq (\sum_{t \leq b} \hat{p}_t) \cdot (T - b) \geq \frac{1}{3}(T - b)$.

For $\sum_{b < t \leq K} \hat{p}_t \geq \frac{1}{3}$, if $T \geq b$, then our $ALG$ is optimal and $\text{diff}_T$ is zero. If $T < b$, $c(ALG_T) = b$ and $c(OPT_T) = T$, then $\text{diff}_T = b - T$. Thus, since $T < b \leq K$, we have $\text{diff}_T = b - T \leq O(EMD(\hat{p}, T))$ because $EMD(\hat{p}, T) \geq (\sum_{b < t \leq K} \hat{p}_t) \cdot (b - T) \geq \frac{1}{3}(b - T)$. $\qquad\square$

Next, we consider the case where $K < b$. Intuitively, a significant amount of mass must be in the $K$-prefix to make the optimal policy $A_K$ choose to rent for a while and then buy on day $K + 1$. The following observation illustrates the minimum amount of mass that needs to be in the $K$-prefix.

**Observation 2.** Assume that $A_K$ is optimal. In the proof of Lemma 20, by comparing the cost between $A_K$ and $A_0$, we have $K \leq \sum_{t \leq K}(K + b - t)\hat{p}_t \leq (K + b)\sum_{t \leq K} \hat{p}_t$. This implies $\sum_{t \leq K} \hat{p}_t \geq \frac{K}{K+b}$.

The next lemma considers the case where $K < b$ and there is enough mass in the $K$-prefix.

**Lemma 21.** *Assume that $A_K$ is the optimal policy for $\hat{p}$ where $K < b$ and $\sum_{t \leq K} \hat{p}_t \geq 0.025$. Taking $ALG = A_\infty$, we have $\text{diff}_T = c(ALG_T) - c(OPT_T) \leq O(EMD(\hat{p}, T))$.*

*Proof.* Let $ALG = A_\infty$. If $T \leq b$, then our $ALG$ is optimal and $\text{diff}_T$ is zero. If $T > b$, $c(ALG_T) = T$ and $c(OPT_T) = b$, then $\text{diff}_T = T - b$. Thus, we have $\text{diff}_T = T - b \leq O(EMD(\hat{p}, T))$ because $EMD(\hat{p}, T) \geq 0.025 \cdot (T - K) \geq 0.025 \cdot (T - b)$. $\qquad\square$

Finally, we prove the case in our algorithm when there is not enough mass in the $K$-prefix.

**Lemma 22.** *Assume that $A_K$ is the optimal policy for $\hat{p}$ where $\sum_{t \leq K} \hat{p}_t < 0.025$. Taking $ALG = A_K$, we have $\text{diff}_T = c(ALG_T) - c(OPT_T) \leq O(EMD(\hat{p}, T))$.*

*Proof.* Since $\sum_{t \leq K} \hat{p}_t < 0.025$, by Observation 2, we know that $K$ cannot be too large. In particular, $K < b/2$.

If $T \leq K$, our algorithm $ALG = A_K$ is optimal.

If $K < T < b/2$, $c(ALG_T) = K + b$ and $c(OPT_T) = T$, then $\text{diff}_T = K + b - T$. If $\sum_{t > b} \hat{p}_t \geq 0.025$, then we have $\text{diff}_T = K + b - T \leq b \leq O(EMD(\hat{p}, T))$ because

$EMD(\hat{p}, T) \geq 0.025 \cdot (b - T) \geq 0.025 \cdot \frac{b}{2}$. Hence, we can assume that both $\sum_{t \leq K} \hat{p}_t < 0.025$ and $\sum_{t \geq b} \hat{p}_t < 0.025$. If $\sum_{\frac{3}{4}b < t < b} \hat{p}_t \geq 0.2$, then $\text{diff}_T \leq b \leq O(EMD(\hat{p}, T))$ because $EMD(\hat{p}, T) \geq 0.02 \cdot (\frac{3}{4}b - T) \geq 0.02 \cdot (\frac{3}{4}b - \frac{1}{2}b)$, and we are done. So, suppose that $\sum_{\frac{3}{4}b < t < b} \hat{p}_t < 0.2$. Then $\sum_{K < t \leq \frac{3}{4}b} \hat{p}_t > 1 - 0.2 - 2 \cdot 0.025 = 0.75$. We will show that the mass in the $\frac{3}{4}b$-prefix is enough to make $A_{\frac{3}{4}b}$ a better policy than $A_K$. In fact,

$$
\begin{aligned}
&c\left(A_{\frac{3}{4}b}\right) - c(A_K) \\
&= \Big[ \sum_{t \leq \frac{3}{4}b} t\hat{p}_t + \sum_{t > \frac{3}{4}b} \left(\tfrac{3}{4}b + b\right)\hat{p}_t \Big] \\
&\quad - \Big[ \sum_{t \leq K} t\hat{p}_t + \sum_{t > K} (K + b)\hat{p}_t \Big] \\
&= \sum_{t > \frac{3}{4}b} \left(\tfrac{3}{4}b - K\right)\hat{p}_t - \sum_{K < t \leq \frac{3}{4}b} (K + b - t)\hat{p}_t \\
&< \left(\tfrac{3}{4}b - K\right) \sum_{t > \frac{3}{4}b} \hat{p}_t - \left(\tfrac{1}{4}b + K\right) \sum_{K < t \leq \frac{3}{4}b} \hat{p}_t \\
&< \left(\tfrac{3}{4}b - K\right) \cdot 0.25 - \left(\tfrac{1}{4}b + K\right) \cdot 0.75 \; < \; 0.
\end{aligned}
$$

which contradicts with the fact that $A_K$ is the optimal policy.

If $b/2 \leq T < b$, $\text{diff}_T = K + b - T$. Note that by Observation 2 and the fact that the mass in the $K$-prefix is less than 0.025, we have in particular $K < \frac{b}{8}$. First, $K \leq O(EMD(\hat{p}, T))$, since

$$
3EMD(\hat{p}, T) \geq 3(T - K)\sum_{t \leq K} \hat{p}_t \geq \frac{3(T - K)}{b + K} \cdot K \geq K
$$

where the last inequality holds because $b + 4K < \frac{3}{2}b \leq 3T$. Hence, we only need to show that $b - T \leq O(EMD(\hat{p}, T))$. Since $c(A_K) \leq c(A_\infty)$, we have

$$
b \leq \sum_t t\hat{p}_t + \big(b - c(A_K)\big)
$$

$$
\Longleftrightarrow 0 \leq \sum_{t < T}(t - b)\hat{p}_t + \sum_{t \geq T}(t - b)\hat{p}_t + \big(b - c(A_K)\big)
$$

$$
\Longleftrightarrow \sum_{t < T}(b - t)\hat{p}_t \leq \sum_{t \geq T}(t - b)\hat{p}_t + \big(b - c(A_K)\big).
$$

Since $b - T < b - t$ for $t < T$, it implies that

$$
(b - T)\sum_{t < T} \hat{p}_t \leq \sum_{t \geq T}(t - b)\hat{p}_t + \big(b - c(A_K)\big)
$$

$$
\Longleftrightarrow b - T \leq \sum_{t \geq T}(t - T)\hat{p}_t + \big(b - c(A_K)\big).
$$

If we can show that $b - c(A_K) \leq 3\sum_{t \leq K}(T-t)\hat{p}_t$, then $b - T \leq O(EMD(\hat{p}, T))$ since $b - \bar{T} \leq 3\sum_{t \geq T}(t - T)\hat{p}_t + 3\sum_{t \leq K}(T-t)\hat{p}_t \leq 3EMD(\hat{p}, T)$, we are done. This holds because

$$b - c(A_K) = \sum_{t \leq K}(K + b - t)\hat{p}_t - K \leq \sum_{t \leq K}(K + b)\hat{p}_t$$

$$\leq \sum_{t \leq K}(3T - 3K)\hat{p}_t \leq 3\sum_{t \leq K}(T - t)\hat{p}_t.$$

where the second inequality holds since $4K + b \leq 3T$ when $T \geq \frac{b}{2}$.

If $T \geq b$, $c(ALG_T) = K + b$ and $c(OPT_T) = b$, then $\text{diff}_T = K$. Again by Observation 2, $EMD(\hat{p}, T) \geq \frac{K}{b+K}(T - K)$, which implies that $2EMD(\hat{p}, T) \geq \frac{2(T-K)}{b+K}K > K = \text{diff}_T$, where the last inequality holds since $2T > b + 3K$. Thus, $\text{diff}_T \leq O(EMD(\hat{p}, T))$.

This proves Lemma 22. □

We can now prove Theorem 18.

*Proof of Theorem 18.* Theorem 18 follows directly from the combination of Lemma 19 and Lemmas 20, 21, and 22. □

One can similarly use the same idea to develop a $\lambda$-robust version for Algorithm 4 as we did in Algorithm 2. Since this is not the main part of the paper, we omit the proof here.

## E. Further Experiment

In Section 2.2, we considered a small synthetic experiment in which both the true distribution and the predicted distribution are bimodal. Here we consider a smoother version of that example by replacing the bimodal prediction with two Gaussian components centered at the modes of the truth. The result also shows that our distributional-prediction algorithm (Algorithm 1) has better performance than the point-prediction algorithm in this more general setting.

The true distribution is the same as in Section 2.2: the buying cost is $b = 100$, and $p_{30} = p_{300} = 1/2$. For the predicted distribution, we use a discretized Gaussian mixture $\hat{p}^{(\sigma)} = \frac{1}{2}\mathcal{N}(30, \sigma^2) + \frac{1}{2}\mathcal{N}(300, \sigma^2)$ normalized over integers. The parameter $\sigma$ controls how spread out each predicted mode is, where we run $\sigma = 0, 5, 10, 15$ in the following experiment. Algorithm 1 uses the full predicted distribution $\hat{p}^{(\sigma)}$ to compute its delayed buying threshold. The point-prediction algorithm samples a single point $\hat{T} \sim \hat{p}^{(\sigma)}$ and then runs the point-prediction rule of Purohit et al. (2018), which buys immediately if $\hat{T} \geq b$ and otherwise keeps renting. For each value of $\sigma$, we ran 200,000 Monte Carlo trials, drawing the true season length $T \sim p$ and, for

*Table 2.* Gaussian-mixture version of the bimodal experiment with $b = 100$ and true distribution $p_{30} = p_{300} = 1/2$. The predicted distribution is a discretized two-component Gaussian mixture with equal weights and centers at 30 and 300. Costs of the algorithms are Monte Carlo averages over 200,000 trials. For reference, the optimal policy under the true distribution $p$ has an expected cost 80.0 (theoretical benchmark).

| $\sigma$ | Our Cost | Point-Prediction Cost | $OPT_p$ |
|---|---|---|---|
| 0 | 85.0336 | 132.6192 | 80.0 |
| 5 | 90.0960 | 132.1461 | 80.0 |
| 10 | 93.0013 | 132.5454 | 80.0 |
| 15 | 94.7367 | 132.3425 | 80.0 |

the point-prediction algorithm, independently drawing $\hat{T} \sim \hat{p}^{(\sigma)}$, as we did in Section 2.2. The resulting average costs are shown in Table 2. This illustrates that the advantage of distributional predictions is not limited to the case where the predicted distributions are bimodal. As a result, our algorithm in Theorem 1 is significantly better than sampling from the prediction and then applying the point-prediction method, and is much closer to the exact optimal policy cost for the true distribution.

## F. Proof of the Centroid Lower Bound of the EMD

Here is a proof for the centroid lower bound of the EMD for sake of completeness. One can also find a similar proof in (Cohen & Guibas, 1997).

**Theorem 23** (Centroid lower bound for Earth Mover's Distance)**.** *Let $p$ and $\hat{p}$ be two probability distributions supported on $\{1, 2, \dots, N\}$, i.e., $\sum_{t=1}^{N} p_t = \sum_{t=1}^{N} \hat{p}_t = 1$. Then we have*

$$\left| \sum_{t=1}^{N} t\, p_t - \sum_{t=1}^{N} t\, \hat{p}_t \right| \leq EMD(p, \hat{p}).$$

*Proof.* Consider any feasible transportation plan $F = (f_{ij})_{i,j \in [N]}$ from $p$ to $\hat{p}$, i.e., $f_{ij} \geq 0$ for all $i, j$, and

$$\sum_{j=1}^{N} f_{ij} = p_i \quad \text{and} \quad \sum_{i=1}^{N} f_{ij} = \hat{p}_j.$$

We can rewrite the difference of centroids as

$$\sum_{i=1}^{N} i\, p_i - \sum_{j=1}^{N} j\, \hat{p}_j = \sum_{i=1}^{N}\sum_{j=1}^{N} f_{ij}\, i - \sum_{j=1}^{N}\sum_{i=1}^{N} f_{ij}\, j$$

$$= \sum_{i=1}^{N}\sum_{j=1}^{N} f_{ij}(i - j).$$

Taking absolute values and applying the triangle inequality,

$$\left| \sum_{i=1}^{N} \sum_{j=1}^{N} f_{ij}(i-j) \right| \leq \sum_{i=1}^{N} \sum_{j=1}^{N} f_{ij} \, |i-j|.$$

Since this holds for any feasible transportation plan $F$, it in particular holds for the optimal plan achieving $EMD(p, \hat{p})$. Thus,

$$\left| \sum_{i=1}^{N} i \, p_i - \sum_{j=1}^{N} j \, \hat{p}_j \right| \leq RHS = EMD(p, \hat{p}),$$

which completes the proof. $\qquad\qquad\square$

