# OpenReview forum: "Ski Rental with Distributional Predictions of Unknown Quality"
_ICML.cc/2026/Conference — ICML 2026 regular_

### Official Review · Reviewer_ZrfF · 2026-03-08

**Soundness:** 3
**Presentation:** 3
**Significance:** 4
**Originality:** 4
**Overall Recommendation:** 5
**Confidence:** 4

**Summary:**

This paper studies ski rental problems within the learning-augmented framework, focusing on distributional predictions rather than point predictions. The paper considers a setting where the true number of ski days $T$ is drawn from an unknown distribution $p$, and the algorithm receives a possibly inaccurate predicted distribution $\hat{p}$. The main contribution is an algorithm that achieves expected additive loss of $O(\min\{\sqrt{b} \cdot \max(\eta, 1), b \log b\})$, where  $\eta = \text{EMD}(\hat{p}, p)$ is the Earth Mover's Distance between the predicted and true distributions, and $b$ is the buying cost. The algorithms has a consistency of $O(\sqrt{b})$ and robustness $O(b \log b)$. Moreover, the authors show that the consistency-robustness tradeoff is Pareto-optimal.

**Compliance With Llm Reviewing Policy:**

Affirmed.

**Key Questions For Authors:**

1. Is it possible to easily obtain an upper bound on the cost of the algorithm that uses a sample of $\hat{p}$ as point prediction then follows the classical approach of Purohit et al.? If not can the authors provide some intuition on how big is the gap between the two?
2. How difficult is it to generalize the algorithm and results to the setting of multiple predictions?
3. Can some algorithmic ideas be generalized or used in other online algorithms with distributional predictions beyond Ski-rental?

**Limitations:**

yes

**Strengths And Weaknesses:**

### Strengths
1. The paper contributes to the literature of algorithms with distributional advice, which is an interesting and growing research sub-area in learning-augmented algorithms.
2. The proposed algorithm is interesting and its analysis are non trivial
3. The authors prove that the algorithm is Pareto-optimal for consistency-robustness, which is a strong result

### Weaknesses
I didn't find any major weaknesses in the paper. Below are some minor ones
1. While the theoretical results are strong, the paper could benefit from experiments comparing the proposed algorithm to the existing algorithms (for example using the mean, median, or sample from the predicted distribution as point distribution)
2. The paper focuses only on the ski-rental problem, which is limited

---

> ### Author Rebuttal · Authors · 2026-03-30
>
> We thank the reviewer for their valuable time and comments!
>
> > Is it possible to easily obtain an upper bound on the cost of the algorithm that uses a sample of $\hat p$ as point prediction then follows the classical approach of Purohit et al.? If not can the authors provide some intuition on how big is the gap between the two?
>
> The “sample then use Purohit et al” approach can work quite poorly in general.  For example, consider the bimodal distribution $p$ where $p_1=1/2$ and $p_M=1/2$ with $M>>b$, and suppose the prediction is perfect, i.e., $\hat p = p$.  The optimal policy is to rent on day 1 and then buy on day 2, and has expected cost essentially $\frac{1}{2} b + 1$.  In this case, our algorithm takes advantage of the full distributional information and achieves a small additive loss $O(\sqrt{b})$. In contrast, sampling a point from $\hat p$ collapses the prediction to one mode and ignores the other.  So, in particular, if we sample the point $M$ but the truth is $1$ then Purohit et al will immediately buy at a cost of $b$, for an additive loss of essentially $b/2$. Similarly, if we sample $1$ and the truth is $M$, then Purohit et al (as written) will just rent until time $M$ and have cost $M$ (although presumably one would robustify this to get cost at most $b$, for an additive loss of $\Omega(b)$).  Thus, in the genuinely distributional setting, there can be a huge gap between the point prediction method and our distributional prediction method that directly uses the full predicted distribution.
>
> There is one setting where the “sample then use Purohit et al” approach works quite well: if the truth $p$ is actually a point mass, then sampling a point from $\hat p$ and applying the point prediction algorithm of Purohit et al. actually gives additive loss of $O(EMD(p, \hat p))$.  This is even better than our bounds, but of course depends on the truth being a fixed (though unknown) value rather than allowing it to be a distribution.
>
> > How difficult is it to generalize the algorithm and results to the setting of multiple predictions?
>
> This is a very interesting extension. Our current paper focuses on the foundational setting of a single distributional prediction. Extending this to multiple predictions seems plausible, and an important first step would be to define the right benchmark: for example, a natural goal would be for the algorithm to perform nearly as good as if it were given the best one among the provided distributional predictions. Some ingredients of our approach, such as reasoning directly at the distribution level and using tail based thresholding, appear compatible with such an extension, although developing this setting cleanly and providing guarantees for it would require additional analysis. We therefore view this as an interesting direction for future work.
>
> > Can some algorithmic ideas be generalized or used in other online algorithms with distributional predictions beyond Ski-rental?
>
> We believe some of the ideas in our paper may be useful beyond the classical ski rental setting, although likely in a problem-specific rather than black-box way. Ski rental is the canonical rent or buy problem, so it is a natural setting in which to study distributional predictions. In particular, some ingredients of our approach seem likely to extend to ski rental variants and related rent or buy problems, such as multi-slope ski rental. For example, the idea of using a controlled delay to balance consistency and robustness is quite intuitive in rent or buy settings. In addition, when the prediction error is measured by the earth mover’s distance, our mass-moving arguments through an optimal transport plan may also be helpful in other settings where the prediction and truth are all distributions over reasonable domains. But for online problems with very different structures, we would expect the specific algorithm to be quite different. So our view is that the broader distributional perspective and some proof ideas may generalize, but extending them to other online problems would likely require new problem-specific algorithmic design.
>
> > While the theoretical results are strong, the paper could benefit from experiments comparing the proposed algorithm to the existing algorithms (for example using the mean, median, or sample from the predicted distribution as point distribution)
>
> While we view this paper as theoretical and so did not include experiments, we point the reviewer to our response to Reviewer ERQT, where we gave some very simple experiments comparing to a sample.

---

> > ### Author Rebuttal · Reviewer_ZrfF · 2026-04-01
> >
> > I thank the authors for addressing my questions. Regarding the experiments, I agree that their absence is not a weakness given that this is a theory paper with a solid and well-presented contribution. However, including some experimental results would strengthen the work. I encourage the authors to add experiments (for e.g. the ones in response to Reviewer ERQT), either in the main paper or in the appendix, to demonstrate the practical relevance of their algorithm beyond the theoretical analysis.
> > I maintain my recommendation for accepting the paper.

---

> > > ### Author Response · Authors · 2026-04-04
> > >
> > > Thank you!  We agree that while the focus of the paper is theoretical and so experiments are not the main point, it makes sense to add some limited experiments like the ones we did for this rebuttal.  We will do so for the final version (most likely in the appendix).

---

### Official Review · Reviewer_uo2E · 2026-03-12

**Soundness:** 3
**Presentation:** 3
**Significance:** 2
**Originality:** 3
**Overall Recommendation:** 4
**Confidence:** 3

**Summary:**

The authors study a classical “Ski Rental” problem from the online algorithms literature, assuming that the algorithm is given access to distributional predictions. In the classical Ski Rental problem, every day the skier makes a decision to either rent skis (cost 1) or buy skis (cost $b > 1$). Without access to predictions, the best known competitive ratio (2) is achieved by renting for $b-1$ days and then buying on the $b$-th day. This problem has since been studied in the setting where the algorithm is given a prediction $\hat T$ for the unknown time horizon, and previous algorithms in this setting perform significantly better when the prediction $\hat T$ is accurate and recover the worst-case performance when $\hat T$ is inaccurate. This paper considers a generalization of this model: the (unknown) time horizon $T$ is drawn from some distribution $p$ and we are given a predicted distribution $\hat p$. This problem has been studied by several works over the last few years, but all prior algorithms assume that the algorithm knows the distance between $p$ and $\hat p$ in advance. As such, an open problem was to design algorithms with distributional predictions which do not require knowledge of this distance.

This work answers this open question and shows an algorithm with additive loss $O(min(\sqrt{b} \cdot max(EMD(\hat p, p), 1), b \log b))$ compared to the optimal policy.

**Compliance With Llm Reviewing Policy:**

Affirmed.

**Final Justification:**

The author's rebuttal was very helpful to clarify the guarantees of their algorithm. Although I still feel that this problem is not very realistic in practice, I believe it is interesting from a theoretical perspective. For these reasons, I have decided to raise my score to Weak Accept.

**Key Questions For Authors:**

- What are the best known guarantees for algorithms which take a fixed point $\hat T$ as a prediction? Can you give more motivation or examples of settings where distributional predictions are useful (as opposed to receiving a prediction $\hat T$ which is sampled from some unknown distribution)?

**Limitations:**

Yes.

**Strengths And Weaknesses:**

Strengths:
- The paper answers a known open question in the literature for ski rental with distributional predictions.
- The writing is clear and easy to follow.
- All mathematical claims are supported with rigorous proofs.

Weaknesses:
- While this is an interesting theoretical problem, I'm not sure that the setting is very applicable or representative in practice.
- The benchmark is not the standard competitive ratio, but rather an additive loss compared to the best policy. In the worst case, if the distribution $\hat p$ is very far from $p$, then the loss is at most an additive $b \log b$, which may be worse than the worst-case in the case without predictions.
- The authors show several lower bounds which all *partially* match the form of the overall upper bound, but do not show a single lower bound which combines all of the factors which appeared in the additive term of the upper bound.

---

> ### Author Rebuttal · Authors · 2026-03-30
>
> We thank the reviewer for their valuable time and comments!
>
> > The setting is [not] very applicable or representative in practice.
>
> We agree that our work is theoretical rather than application-specific. However, ski rental is *the* classical rent-or-buy problem.  It is the simplest setting in which the rent-or-buy dilemma is cleanly exposed, justifying the huge literature on such an “unrealistic” problem.  Moreover, distributional predictions and a distributional truth are also extremely natural, for a number of reasons.  Presumably we are learning from the past so are using something about an empirical distribution, and moreover many ML systems are inherently stochastic, so it makes sense to think of a distributional prediction.  In addition, the true weather and conditions are reasonably modeled as stochastic in the usual ski rental story.  For example, the random event of an early season rain-and-freeze might induce a bimodal distribution in quality ski days (a familiar situation for many North American east coast skiers).  More seriously, in many more realistic situations modeled by ski rental (e.g., snoopy caching or TCP acknowledgement) it also makes sense to think of the truth as being stochastic.  So we view understanding how distributional predictions can be utilized for ski rental as an important and fundamental contribution, even if it is not directly applicable to any particular “realistic” problem.  For more discussion of the importance of ski rental and the motivation for distributional predictions, see Section 1 of the paper.
>
> > Additive loss rather than competitive ratio, and additive $b \log b$ may be worse than the worst-case without predictions.
>
> Both additive loss guarantees and competitive ratio guarantees have been studied in prior work on ski rental, so our use of additive loss is standard for this problem.  Our results are directly comparable to the most relevant previous work (Purohit et al (2018)) which gives additive loss bounds, and does not even have the robustness that we get “for free”. Moreover, our “further robustification” results are essentially competitive ratio results: simply divide both sides by $OPT_p$ in Theorem 2, or see this done explicitly in Corollary 13 in Appendix A.  Finally, the concern about the $O(b \log b)$ term in the worst case is exactly why we introduce the further robustification step: see Theorem 2, the discussion starting on line 174, and Appendix A.  Thus, both issues raised here are addressed in the paper, and in the final version we will make the role of the Appendix A further robustification more explicit in the main text.
>
> > No single matching lower bound
>
> We agree that we do not currently state a single lower bound theorem that combines all parts of the upper bound into one formula. However, the lower bounds do match the upper bound in several important ways. We would point to the review of Reviewer QvXD, who points out our different lower bounds as a *strength* despite the fact that they do not form a single clean bound.  As they say: “Since the upper bound on the algorithm's performance has an entangled dependence both on distributional distance and on the cost parameter $b$, it is not easy to establish its tightness via any single lower bounding approach, but it is possible to provide several bounds that cover/disallow possible improvement pathways, and it is convincingly done here: a Pareto property, worst-case robustness, the multiplicative nature of the best-case performance dependence jointly on the distance and on parameter $b$, etc., are all established via different instances/techniques.”
>
> > Known guarantees for algorithms which take a fixed point $\hat T$ as a prediction, and examples where distributional predictions are useful rather than sampling a point prediction?
>
> For the single point prediction version of ski rental, the best known guarantees are given by Purohit et al. They show additive loss that is linear in the distance between the prediction and the truth value, i.e., in $|\hat T - T|$ (where $T$ is the true value). They also give a robustified bound similar to our further robustification.
>
> As for sampling a point from a predicted distribution and then applying the point prediction algorithm of Purohit et al., this has bad performance even on simple distributions like bimodals.  In particular, when the distributional prediction is reasonably accurate, our approach will have additive loss on the the order of $O(\sqrt{b})$ even for bimodal settings, while sampling and then using Purohit et al.’s algorithm will have additive loss $\Omega(b)$.
>
> Reviewer ERQT asked for a simple empirical study comparing to the point prediction baseline, so we have implemented exactly this approach for a bimodal distribution. We refer this reviewer to our response to Reviewer ERQT to see that empirically our approach is much better on bimodal instances, as well as our response to Reviewer ZrfF for some calculations.

---

> > ### Author Rebuttal · Reviewer_uo2E · 2026-04-03
> >
> > Thank you for your response to my questions! This was very helpful to clarify the guarantees of your algorithm. Although I still feel that this problem is not very realistic in practice, I think it is interesting from a theoretical perspective. For these reasons, I will raise my score.

---

> > > ### Author Response · Authors · 2026-04-04
> > >
> > > Thank you!  We appreciate your open mind and attention to our rebuttal.

---

### Official Review · Reviewer_QvXD · 2026-03-13

**Soundness:** 3
**Presentation:** 2
**Significance:** 3
**Originality:** 3
**Overall Recommendation:** 4
**Confidence:** 4

**Summary:**

The paper proposes a set of novel bounds for the classic ski rental problem with predictions. As opposed to the vast majority of algorithms-with-predictions works so far, the authors assume that predictions and the ground truth are both distributional in nature (i.e. the unknown time horizon T is distributed over the integers within some arbitrary finite range, and there is a predicted distribution, whose quality is unknown). In contrast to some prior work, which assumed that there is a known upper bound on the distance of the predicted distribution from the true distribution, this paper develops an algorithm and bounds that do not require such knowledge.

In fact, the proposed algorithm achieves better and better performance if the predicted distribution is accurate, while also offering a robust upper bound in the case that the predicted distribution is misguided. Furthermore, to complement the upper bound, the authors propose assorted lower bounds, showing that the developed upper bound can be thought of as tight in several ways.

**Compliance With Llm Reviewing Policy:**

Affirmed.

**Final Justification:**

The rebuttal, as well as the rest of the discussion (rebuttals to other reviewers) reinforced my prior assessment, which I stand by. I believe the paper contains useful and non-trivial theory insights, and while it had lacked evaluations that some of the reviewers have suggested, I think the paper is well-positioned and valuable as a theory paper even without further empirics.

I still think the paper could be pushed further in the direction of generalizing the proposed methods beyond ski rental, as well as that the presentation of the technical details (such as the proof) could be improved --- and the authors have been receptive to these concerns/suggestions and have in particular promised to improve the exposition for clarity. Hence, I keep my positive evaluation.

**Key Questions For Authors:**

N/A (but see above in the Strengths and Weaknesses section and feel free to address any points there).

**Limitations:**

yes

**Strengths And Weaknesses:**

Strengths:

+ The paper develops a simple and intuitive algorithm for the classical ski rental problem that for the first time can leverage distributional predictions without assuming any a-priori goodness on them. Aside from a few exceptions (discussed in the paper), other work in this area has mostly focused on point predictions, and this paper may therefore serve as a beacon for a substantial amount of future algs-with-distributional-predictions research.

+ Furthermore, the authors prove a fairly tight (in some senses) bound on the algorithm's performance, which establishes both robust behavior when predictions are bad, and improved behavior when the predictions are good --- and the proof itself is overall quite nontrivial and contributes novel techniques to the literature, including a possibly fruitful argument scheme based on optimal transport (but see below).

+ The collection of lower bounds provided is quite nice. Since the upper bound on the algorithm's performance has an entangled dependence both on distributional distance and on the cost parameter b, it is not easy to establish its tightness via any single lower bounding approach, but it is possible to provide several bounds that cover/disallow possible improvement pathways, and it is convincingly done here: a Pareto property, worst-case robustness, the multiuplicative nature of the best-case performance dependence jointly on the distance and on parameter b, etc., are all established via different instances/techniques.

Weaknesses:

- Having studied the provided proof of the algorithm's performance, it appeared to me that there is an unused opportunity to extend the proposed optimal transport-based approach to many other distributional-predictions settings beside ski rental; in that sense, I wish the paper had expanded its reach beyond its present setting, and showcased the application of these techniques more broadly. It is of course true that lower bounds may not be as transferrable across settings, but the upper bound part sounds like it could. If a more comprehensive version of this paper was provided, it would further increase its significance to the subfield --- in a similar way as the robust-formulation Besbes et al. paper discussed has enhanced the message it conveys via broader coverage of algs-with-predictions settings (with respect to their approach).

- While I found it nice that various proof aspects made it into the main part of the manuscript, making it more self-contained to review and allowing one to verify some important construction details, I found the proof sketches overall quite messy, with lots of inline math not all of which may have merited inclusion into the main part. I would go lighter on notation/equations and dedicate more space to verbal, higher-level explanations --- I found myself venturing into the appendix for intuition/verification several times, so I believe refocusing the main/appendix split should be possible and beneficial.

---

> ### Author Rebuttal · Authors · 2026-03-30
>
> We thank the reviewer for their valuable time and comments!
>
>
> > Having studied the provided proof of the algorithm's performance, it appeared to me that there is an unused opportunity to extend the proposed optimal transport-based approach to many other distributional-predictions settings beside ski rental; in that sense, I wish the paper had expanded its reach beyond its present setting, and showcased the application of these techniques more broadly. It is of course true that lower bounds may not be as transferrable across settings, but the upper bound part sounds like it could. If a more comprehensive version of this paper was provided, it would further increase its significance to the subfield --- in a similar way as the robust-formulation Besbes et al. paper discussed has enhanced the message it conveys via broader coverage of algs-with-predictions settings (with respect to their approach).
>
> We appreciate this suggestion and agree that it would be very interesting to extend the optimal transport-based approach to distributional prediction settings beyond ski rental. Our goal in this paper, however, is not to develop a general optimal transport framework for many learning augmented online problems, but rather to study the distributional prediction viewpoint in one of the most canonical online problems. We chose ski rental because it is a central problem in the online algorithms literature as well as the algorithms-with-predictions literature, and it lets us study, in a simple setting, what happens when the prediction is a distribution rather than a single value. In that sense, we view this paper as a focused study of one important problem, rather than as developing a broad framework for many problems.  We note that this is similar to the paper (Dinitz et al 2024) we cite in our related work section (Section 2.3), which gave a focused study of distributional predictions on a specific problem (binary search trees) that is particularly important in one area (data structures). But we certainly hope that our paper will both help inspire others to think about distributional predictions rather than point predictions, and to consider EMD and optimal-transport based techniques.
>
> We do believe that some ingredients of our approach seem likely to extend to ski rental variants and related rent or buy problems, such as multi-slope ski rental. For example, the idea of using a controlled delay to balance consistency and robustness is quite intuitive in rent or buy settings. In addition, when the prediction error is measured by the earth mover’s distance, our mass-moving arguments through an optimal transport plan may also be helpful in other settings where the prediction and truth are all distributions over reasonable domains. But for online problems with very different structures, we would expect the specific algorithm to be quite different. So our view is that the broader distributional perspective and some proof ideas may generalize, but extending them to other online problems would likely require new problem-specific algorithmic design.
>
> Additionally, we believe the right error metric is problem-dependent: in our setting, earth mover’s distance is especially natural because it captures how far probability mass is shifted, whereas in other problems different metrics (e.g., KL divergence) may be more appropriate. So while we hope this paper helps start a broader line of work on distributional predictions, the contribution here is intentionally centered on ski rental (where the natural error metric is EMD) and on showing that the distributional viewpoint already leads to new algorithms, guarantees, and lower bound phenomena even in this classical online problem.
>
>
> > While I found it nice that various proof aspects made it into the main part of the manuscript, making it more self-contained to review and allowing one to verify some important construction details, I found the proof sketches overall quite messy, with lots of inline math not all of which may have merited inclusion into the main part. I would go lighter on notation/equations and dedicate more space to verbal, higher-level explanations --- I found myself venturing into the appendix for intuition/verification several times, so I believe refocusing the main/appendix split should be possible and beneficial.
>
> We appreciate this comment. We wanted to keep the paper reasonably self-contained while still presenting the main ideas in the main body of the paper, but we agree with the comment and will revise the presentation for the final version.  In particular, we will ensure that the main body places more emphasis on the high-level explanation and overall structure, with less inline math, and keeps only the most important formulas and constructions.  We agree with the reviewer that this will make the paper easier to read while preserving the technical clarity of the current version.

---

> > ### Author Rebuttal · Reviewer_QvXD · 2026-04-04
> >
> > I thank the authors for carefully responding to various points raised in my review, and to those of the other reviewers. In particular, I welcome the brief remarks --- in response to my question about possible generalizations of the presented techniques to other algs-with-predictions settings --- on how the "right" metric for the task can influence what techniques are appropriate; I would encourage follow-up work that would in particular explore which tasks are compatible with EMD (and hence presumably with the optimal transport techniques discussed here).
> >
> >
> > Additionally, it was interesting to see the brief extra experimental results provided in response to other reviewers, and the brief empirical remarks on how bimodality etc. can highlight the advantages of using distributional predictions over point predictions. Overall, my opinion on this paper remains positive, and I will keep my positive score and continue to be in favor of acceptance.

---

> > > ### Author Response · Authors · 2026-04-04
> > >
> > > Thank you! We agree that followup work exploring other settings where EMD is a particularly natural error metric would be quite interesting.

---

### Official Review · Reviewer_ERQT · 2026-03-13

**Soundness:** 3
**Presentation:** 3
**Significance:** 4
**Originality:** 4
**Overall Recommendation:** 5
**Confidence:** 4

**Summary:**

This paper considers Ski rental under "algorithms with predictions" with *distributional* predictions and *unknown* prediction error. The authors propose a novel algorithm that achieves $\mathcal{O}(\min(\max(\text{EMD}(p,\hat{p}),1)\sqrt{b}, b\log b))$ additive loss ($\mathcal{O}(\sqrt{b})$ consistency, $\mathcal{O}(b\log b)$ robustness). Tight lower bounds and Pareto-optimality of this tradeoff are established. Also, the authors propose a parameterized variant that smoothly interpolates between their prediction-based algorithm and the classical 2-competitive algorithm.

**Compliance With Llm Reviewing Policy:**

Affirmed.

**Final Justification:**

I support the acceptance of this paper due to its technical novelty. While I note the concerns regarding the lack empirical evaluations, the authors' rebuttal includes a simple numerical simulation that narrows this gap. I recommend that the simulation be included in the revised manuscript to provide a more comprehensive evaluation.

**Key Questions For Authors:**

- Could you provide a small empirical study (synthetic or historical data) comparing Algorithm 1 to point-prediction or known-error baselines?
- Brief intuitive explanation in main text for why delay $\sqrt{b}$ is the theoretical sweet spot?
- Do lower-bound hardness results relax for smooth/parametric $p,\hat{p}$ (e.g., Poisson, log-concave)?

**Limitations:**

The paper only includes a generic impact statement and does not explicitly discuss limitations. I suggest that the authors comment on the absence of empirical validation, the fact that the lower bounds rely on adversarial discrete distributions that may be pessimistic for ML-style forecasts, and the computational cost of computing $\hat{K}$ and $U$ when $N$ is large.

**Strengths And Weaknesses:**

**Strengths:** The paper lifts the known-prediction-error assumption for distributional predictions in ski rental, which is novel and resolves an open question. The theoretical analysis is rigorous, relying on optimal transport arguments and a Pareto-optimality result. The overall presentation is clear and well structured.

**Weaknesses:** The paper is purely theoretical and does not include any empirical validation. The presentation lacks an intuitive explanation for why a delay of exactly $\sqrt{b}$ is optimal. The lower bounds rely on adversarial discrete distributions, so it is unclear whether the same tradeoffs hold for more benign parametric or smooth distributions.

---

> ### Author Rebuttal · Authors · 2026-03-30
>
> We thank the reviewer for their valuable time and comments!
>
> > Could you provide a small empirical study (synthetic or historical data) comparing Algorithm 1 to point-prediction or known-error baselines?
>
> As a small empirical study, we compare Algorithm 1 to Purohit et al.'s point prediction algorithm in a bimodal distribution. We choose this distribution because we would like to use a realistic probability distribution but one that is far from being a single point (e.g., a concentrated Gaussian), in order to show the differences between distributional and point predictions. One could imagine that the ski duration can be short or long, depending on factors like whether it rains in January (destroying the snow base) or if the skier gets injured early on. Thus in at least some settings it is natural to consider a bimodal distribution as the truth $p$.
>
> We take the buy cost $b=100$, and let the true distribution $p$ have two modes, with $p_{30} = 1/2$ and $p_{300} = 1/2$.  We then consider a bimodal predicted distribution $\hat p^{(d)}$, defined by $\hat p_{30+d}^{(d)} = 1/2$ and $\hat p^{(d)}_{300-d}=1/2$. For this family, $EMD=d$. For each value of $d$, corresponding to the full predicted distribution $\hat p^{(d)}$, we ran 200,000 Monte Carlo trials, drawing the true season length $T$ from $p$, and then evaluating the cost of Algorithm 1. As for a point prediction, we sample one point $y \sim \hat p^{(d)}$ and run the point prediction algorithm given by Purohit et al. (buy immediately if $y>=b$, and otherwise keep renting) using only that sampled point. The resulting average costs are shown in the table below when $d \in \{0,5,10\}$.
> | EMD(p, hat p^(d)) | Algorithm 1 cost | Point prediction cost | Optimal policy cost |
> | ----- | ----- | ----- | ----- |
> | 0 | 85.1067 | 132.9766 | 80.0 |
> | 5 | 87.3655 | 131.9728 | 80.0 |
> | 10 | 90.1374 | 132.5884 | 80.0 |
>
> The table reports Monte Carlo estimates from 200,000 trials. In this example, the exact expected costs are easy to compute: Algorithm 1 has expected costs 85.0, 87.5, 90.0 for $d=0,5,10$, respectively, while the point prediction algorithm has expected cost 132.5 for all three values of $d$. For reference, the optimal policy under the true distribution $p$ has an expected cost 80.0. This illustrates the regime where distributional predictions are perhaps the most useful: when the prediction is genuinely bimodal, Algorithm 1 is significantly better than sampling from the prediction and then applying the point prediction method.
>
> > Brief intuitive explanation in main text for why delay $\sqrt{b}$ is the theoretical sweet spot?
>
> We agree that a brief intuition for the $\sqrt{b}$ shift would be helpful in the main text, and we will add such a discussion. At first glance, it is not obvious why the right delay should be $\sqrt{b}$ rather than, say, $b^{1/4}$ or $b^{3/4}$. The reason becomes clear in the analysis. If we replace the $\sqrt{b}$ shift by a general delay of $D$, then in our optimal transport argument, the only probability mass that can cause a large change in cost is mass that moves far enough in the transport plan to cross the extra D-day delay we added before buying. By the definition of EMD, the total amount of such mass is bounded by $EMD/D$. Since each unit of such mass can change the rental cost by at most $b$, in our required upper bound the coefficient becomes $b/D$. On the other hand, in the transport plan there is also the part where the moving distance contributes directly; this is bounded by $D$ times the transported mass, and hence by $D$. The choice $D=\sqrt{b}$ then gives the best tradeoff, which is why $\sqrt{b}$ is the theoretical sweet spot in our analysis.
>
> > Do lower-bound hardness results relax for smooth/parametric $p, \hat p$ (e.g., Poisson, log-concave)?
>
> This is an interesting question. In our paper, both the true distribution $p$ and the predicted distribution $\hat p$ are allowed to be arbitrary distributions; we do not assume they belong to a restricted family such as Poisson or log-concave. Our lower bounds are proved in this general setting, using simple bimodal/multimodal constructions. Since these constructions are not restricted to smooth parametric families, our current lower bounds do not directly answer whether the same hardness still holds under such additional assumptions. It is possible (and indeed quite likely) that imposing extra structure on both $p$ and $\hat p$ could lead to sharper guarantees. However, a common goal in the algorithms-with-predictions literature is to obtain guarantees under as few assumptions as possible on the true instance and the prediction, and our paper follows that modeling philosophy. We agree that understanding the smooth/parametric case would be an interesting direction for future work.
>
> Limitations: These are excellent points.  We will revise the final version to include mention of these limitations.

---

> > ### Author Rebuttal · Reviewer_ERQT · 2026-04-04
> >
> > Thank you for your detailed response. I will keep my score.

---

> > > ### Author Response · Authors · 2026-04-04
> > >
> > > Thank you!

---

### Decision · Program_Chairs · 2026-04-30

**Decision:**

Accept (regular)

**Comment:**

This paper revisits ski rental in the setting of algorithms with distributional predictions, where the algorithm receives a predicted distribution over the ski horizon rather than a point estimate. The main result is an algorithm that achieves improved consistency-robustness guarantees, along with Pareto-optimality results and tight lower bounds. Reviewers generally agree the work is technically strong, with rigorous and non-trivial analysis and meaningful contributions to the theory of online algorithms with predictions. On the other hand, concerns are mainly about the lack of empirical evaluation and limited discussion of applicability beyond the ski rental setting. While it is true that experimental validation is not strictly required for purely theoretical contributions, the learning-augmented framework is often motivated by practical settings, and the absence of any empirical evaluation weakens the connection to this motivation and limits evidence of practical relevance.

Overall, despite these limitations, the theoretical contributions are viewed as solid, and the paper sits at a borderline level of acceptance: it is strong enough in technical depth and completeness of results to be considered for ICML, but with noted weaknesses in empirical validation and motivation that limit its overall strength.